# Ultra-narrow donor-acceptor nanoribbons

James Lawrence [1,2] ✉, Luka Đorđević [3,9], Fabienne Bachtiger[1,10], Harry Pinfold[1], Marc Walker [4], Jiong Lu [2,5], Gabriele C. Sosso [1], Davide Bonifazi [3,6] & Giovanni Costantini [1,7,8] ✉

Donor–acceptor (D–A) architectures underpin many high-performance conjugated polymers but remain largely unexplored in atomically precise nanoribbons. Here, we report the on-surface synthesis of ultra-narrow D–A nanoribbons using two complementary brominated precursors based on the electron donor peri-xanthenoxanthene and the acceptor anthanthrone. High-resolution scanning tunnelling microscopy, non-contact atomic force microscopy and scanning tunnelling spectroscopy reveal submolecular structural and electronic features of the resulting nanoribbons. Homopolymerisation of each precursor yields structurally well-defined donor-only and acceptor-only nanoribbons, whose electronic character strengthens with length. Co-deposition of both precursors produces mixed D–A nanoribbons with tuneable electronic structures governed by monomer sequence. The spatial character and energetic alignment of their frontier orbitals match gas-phase density functional theory calculations, while a simplified linear combination of molecular orbitals model captures dominant trends. This bottom-up synthetic strategy enables precise control over nanoribbon composition and functionality, offering a versatile platform for engineering π-conjugated nanostructures with tailored optoelectronic properties.

Confining graphene into one dimensional nanoribbons via bottom-up on-surface synthesis has attracted considerable interest in the past fifteen years[1–3]. Unlike 2D graphene, graphene nanoribbons (GNRs) exhibit a band gap, making them promising semiconductor candidates[4–7]. This band gap can be tuned by varying the length, width and edge structure of graphene nanoribbons[1,8]. Recent advances in on-surface synthesis have enabled the fabrication of atomically precise nanoribbons through the controlled design of molecular precursors, allowing detailed studies of structure–electronic property relationships.

Beyond tuning basic structural parameters, heteroatom doping has also been shown to significantly modify the electronic properties of GNRs. Internal doping with boron[9–11] and nitrogen[12], along with edge functionalisation using groups containing nitrogen[13–18], sulfur[19] and oxygen[20–22], enables GNRs to act as electron donor (p-type) or electron acceptor (n-type) materials. Moreover, increasing the length of a given GNR enhances its donor or acceptor character by reducing the band gap through quantum confinement effects.

Donor–acceptor (D-A) nanoribbons represent a natural extension of these strategies, combining distinct monomer units to enable further modulation of electronic properties. However, such architectures remain largely underexplored in the GNR field. Most reported examples involve junctions between all-carbon segments with different widths or edge structures[23–26], leading to variations in band gap and interesting topological effects[27–30], or the combination of sections incorporating various heteroatoms, which typically exhibit only limited electronic differences[13,31].

[1]Department of Chemistry, University of Warwick, Coventry, UK. [2]Department of Chemistry, National University of Singapore, Singapore, Singapore. [3]School of Chemistry, Cardiff University, Cardiff, UK. [4]Photoemission Research Technology Platform, Department of Physics, University of Warwick, Coventry, UK. [5]Institute for Functional Intelligent Materials, National University of Singapore, Singapore, Singapore. [6]Department of Organic Chemistry, University of Vienna, Vienna, Austria. [7]School of Chemistry, University of Birmingham, Birmingham, UK. [8]School of Physics and Astronomy, University of Birmingham, Birmingham, UK. [9]Present address: Department of Chemical Sciences, University of Padova, Padova, Italy. [10]Present address: Department of Chemical Engineering, University College London, London, UK. ✉e-mail: jmlaw91@nus.edu.sg; g.costantini@bham.ac.uk

In contrast, true D-A architectures—featuring alternating electron-rich (donor, D) and electron-poor (acceptor, A) units—have become a benchmark in high-performance conjugated polymers for organic electronics and photovoltaics[32,33]. Their success is largely due to this design's ability to precisely tune energy levels and narrow the band gap. In particular, the hybridisation of frontier molecular orbitals positions the LUMO of D-A systems close to that of the acceptor and the HOMO near that of the donor, allowing near-independent control over both energy levels (see Fig. S1 and Section 21 in the attached Supplementary Information, SI)[32,34]. Moreover, selecting strong D and A moieties—i.e. with high-lying $HOMO_D$ and low-lying $LUMO_A$—can yield narrow band gaps[35]. These features enhance device performance by reducing charge carrier injection barriers, improving ambient stability, enabling ambipolar transport and maximising light adsorption in the near-infrared region[32,36–38].

In polymer chemistry, D-A conjugated polymers are typically synthesised by preparing D and A monomers separately, followed by polymerisation through transition-metal-catalysed cross-coupling reactions[39]. Here, we adapt this strategy to an on-surface synthetic paradigm, employing two molecular precursors that are strong electron donor and acceptor analogues of anthanthrene (AA, Fig. S2a), to form ultra-narrow D-A nanoribbons. The resulting nanoribbons are effectively fully conjugated D-A ladder polymers[40].

*Peri*-xanthenoxanthene (PXX) and anthanthrone (AO) were employed as D and A units, respectively (Fig. 1). Structurally, PXX is an O-doped, stable analogue of AA (Fig. S2a)[41,42]. The incorporated oxygen atoms are conjugated with the π-system, contributing electron density through resonance and thereby raising the HOMO energy level, which enhances its donor character. PXX has been previously used in flexible organic light-emitting diode (OLED) displays[43,44], with substituted derivatives exhibiting good injection and transport properties as well as high chemical and thermal stability[41,45]. Structural modifications, such as extending the aromatic core or introducing alkyl-imide groups, have been shown to tune its optical and electronic properties, including emission yield and HOMO level[46–48]. Short PXX-based nanoribbons have also been synthesised in solution through a lengthy multi-step procedure, showing progressively elevated HOMO levels with increasing length[47,49,50]. On the contrary, the quinoidal structure of AO withdraws electrons, stabilising the LUMO and conferring acceptor character to the molecule[51–54]. AO can thus be viewed as the natural acceptor analogue of PXX (Fig. S2a). Its brominated derivative—commonly known as Vat Orange 3 (VO3) due to its application as a vat dye[55,56]—has also been explored as an organic semiconductor in OFETs, valued for its charge mobility, low toxicity and biodegradability[57–59]. Additionally, AO derivatives have been employed as hole-transport materials in perovskite solar cells[60].

In the following, brominated derivatives of PXX and AO—$Br_2PXX$ and VO3, respectively—are employed to synthesise polymeric nanoribbons via surface-catalysed Ullmann coupling followed by dehydrogenation[1]. Structural characterisation is carried out using scanning tunnelling microscopy (STM), bond-resolving STM (BR-STM) and non-contact atomic force microscopy (nc-AFM) under ultrahigh vacuum (UHV) conditions, while scanning tunnelling spectroscopy (STS) is used to probe their local electronic properties. We demonstrate the successful formation of all-donor and all-acceptor nanoribbons, in some cases exceeding 20 monomer units in length. Furthermore, we show that the on-surface copolymerisation of PXX and AO enables a systematic modulation of the electronic structure, yielding ultra-narrow donor–acceptor nanoribbons with composition-dependent frontier-state distributions. The electronic structure of these ribbons, as well as the spatial character of their frontier orbitals, is in excellent agreement with gas-phase DFT calculations of the corresponding molecular orbitals. Additionally, a simple linear combination of molecular orbitals (LCMO) model captures the main electronic trends observed across different donor–acceptor compositions. This synthetic strategy demonstrates a viable route for precisely tuning the properties of carbon-based nanoribbons and offers a foundation for achieving application-specific electronic behaviour through controlled subunit composition.

## Results and discussion

Pure donor nanoribbons were synthesised by depositing $Br_2PXX$ via thermal sublimation onto a room temperature Au(111) surface, followed by post-annealing in 50 K increments. A representative STM image of the as-deposited $Br_2PXX$ molecules, forming a characteristic halogen-bonded kagome assembly[61], is shown in Fig. 2a. Figure 2b displays an image acquired after annealing to 523 K, revealing the formation of nanoribbons. X-ray photoelectron spectroscopy (XPS) measurements performed during the nanoribbon growth process (Fig. S3d) indicate the onset of debromination and changes in the carbon bonding environment upon annealing above approximately 410 K, providing supporting evidence for the nanoribbon formation observed by STM. As shown in Fig. S3b, at intermediate annealing temperatures, nanoribbons assemble into compact islands alongside Br atoms released during the Ullmann coupling reaction. Upon further annealing to 523 K, most of the co-adsorbed Br atoms desorb from the surface (Figs. S3c and 2b), as confirmed by XPS (Fig. S3e). This also results in the dispersion of the PXX nanoribbons across the surface, as they do not self-assemble without co-adsorbed bromine atoms. Additional details on the intermediate steps of PXX nanoribbon synthesis are provided in Section 3 of the SI.

Owing to the pro-chirality of the $Br_2PXX$ precursor[61], nanoribbons can form via two distinct coupling pathways: between PXX units of the same pro-chirality, yielding 'straight' junctions (Fig. S4a), or between units of opposite pro-chirality, resulting in 'alternating' junctions (Fig. S4b). The predominance of alternating junctions—causing the wavy morphology of most nanoribbons (Fig. 2c, g)—suggests that the initial relative positions of the reacting C-Br groups play a key role in determining the coupling efficiency. Specifically, the reaction appears to be favoured when the radicals generated by precursor debromination are aligned face-to-face. The same steric and configurational considerations apply to the VO3 precursors during the formation of the acceptor nanoribbons (see below). Further possible reaction mechanisms are discussed in Sections 4, 6 and 9 of the SI.

STM images reveal that many nanoribbons exhibit structural defects, likely caused by contaminants such as unintended regioisomers of $Br_2PXX$ or trisubstituted derivatives that could not be removed[61]. For instance, molecules brominated at the 1-position instead of the 3-position (1 in Fig. S5) tend to form 'straight' rather than the regular 'alternating' junctions, while tribrominated species (2 in Fig. S5) may result in branched structures similar to those observed at the centre of Fig. S6b, highlighted by the red circle. Additional defects, such as kinks, are attributed to the formation of 5-membered rings arising from misoriented monomer coupling (Fig. S6), a phenomenon also observed in other brominated precursors used for nanoribbon synthesis[62].

To determine the structure of the PXX nanoribbons with higher precision, we employed constant-height nc-AFM frequency shift imaging with a CO-functionalised tip (see Methods)[63,64]. A representative image is shown in Fig. 2c, confirming that the synthesised nanoribbons match the expected structure depicted in Fig. 2d. For PXX nanoribbons, nc-AFM proved more effective than BR-STM, as strong tunnelling current modulation in longer ribbons partially obscured their structure in BR-STM images. As discussed in Sections 7 and 16 of the SI, this effect is linked to the electronic structure of the PXX nanoribbons.

Following the characterisation of donor-type nanoribbons, we turned to the synthesis of their acceptor counterparts using the VO3 precursor. When deposited at room temperature on Au(111), VO3 forms compact self-assembled islands spanning both fcc and hcp regions of the herringbone reconstruction (Fig. 2e). The resulting

square lattice is primarily stabilised by type II halogen bonds, as revealed by BR-STM imaging and supported by a structural assignment (Fig. S10).

The polymerisation behaviour of VO3 closely resembles that of Br₂PXX. Annealing to 473 K results in the formation of short nanoribbons that are co-adsorbed alongside bromine atoms. Further annealing to 573 K leads to bromine desorption (Fig. S11 and 2f) and the formation of long nanoribbons. Unlike PXX nanoribbons, the AO nanoribbons tend to assemble into compact domains following bromine desorption. A tentative molecular arrangement for this assembly, based on weak hydrogen bonding between ketone groups and adjacent hydrogens, is presented in Fig. S12.

The structure of the AO nanoribbons is clearly resolved via CO-tip nc-AFM, with ketone groups appearing as sharp, distinct features (Fig. 2g, h)[65]. In many cases, bright junctions are observed connecting the termini of adjacent ribbons. These are likely the result of precursor misalignment during polymerisation, leading to the formation of a single C-C bond rather than a fully fused junction (Fig. S13). Although no distinct polymeric intermediate is observed at lower annealing temperatures, the reasons why such junctions appear more frequently in AO nanoribbons than in their PXX counterparts remain to be clarified.

STS measurements were performed to investigate the impact of length on the electronic structure of both PXX and AO nanoribbons, with a particular focus on their donor/acceptor strength. dI/dV spectra of short nanoribbons ranging from 1 to 4 units are shown in Fig. 3a, f. The most pronounced shifts in frontier energy levels occur in the short-length regime, consistent with gas-phase DFT calculations (Fig. S2b), while changes beyond four units are minimal. In both nanoribbon types, the HOMO shifts upwards and the LUMO downwards with increasing length, resulting in evident narrowing of the band gap. This trend reflects an increase in donor character for longer PXX nanoribbons and is qualitatively reproduced by theory. For PXX, it also aligns well with cyclic voltammetry and photophysical measurements on similar short PXX-derived nanoribbons synthesised in solution[49].

Figure 3 also shows constant height dI/dV images of trimers for both nanoribbon types, acquired with a CO-functionalised tip at voltages corresponding to the frontier resonances. The spatial distributions of these states closely match the corresponding simulated dI/dV images. Additional comparisons—including higher and lower energy resonances and alternative tip terminations—are presented in Fig. S15 and show good agreement with theory too.

For longer PXX nanoribbons, the highest occupied state becomes increasingly difficult to resolve as it approaches the Fermi level, as seen for the tetramer in Fig. 3a. High-resolution dI/dV spectra around 0 V for a set of longer ribbons are shown in Fig. S16, with an example of a 21-unit nanoribbon in Fig. S17. The highest occupied state is found to cross the Fermi level at a length of 5 or 6 units, after which it empties via charge transfer to the surface and stabilises near +40 mV. This charging behaviour may contribute to the reduced tendency of longer PXX nanoribbons to self-assemble, possibly due to repulsive intermolecular interactions. The same state also accounts for the contrast observed in BR-STM at low bias (Fig. S7). Notably, the unoccupied

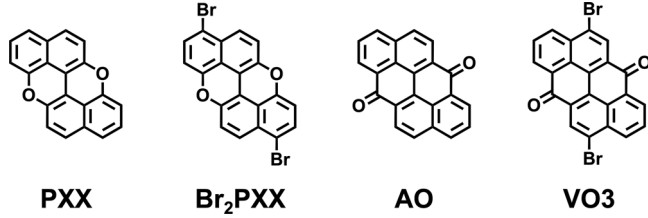

**Fig. 1 | Chemical structures of the donor and acceptor units employed in this study.** Chemical structure of *peri*-xanthenoxanthene (PXX), its brominated derivative (Br₂PXX), anthanthrone (AO) and its brominated derivative (VO3).

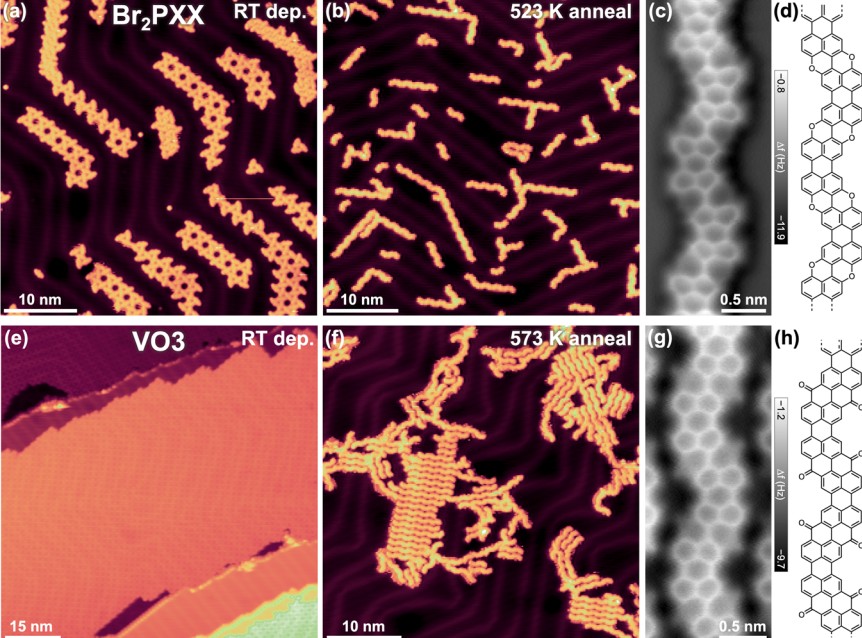

**Fig. 2 | Synthesis of pure PXX and AO nanoribbons on Au(111). a, b** STM images of Br₂PXX after deposition at room temperature and subsequent annealing to 523 K. **c** Constant-height nc-AFM frequency shift image (CO tip) of a PXX nanoribbon; the brighter upper and lower sections of the nanoribbon are related to the position of the ribbon relative to the underlying Au(111) herringbone reconstruction. **d** Corresponding chemical structure. **e, f** STM images of VO3 after deposition at room temperature and subsequent annealing to 573 K. **g** Constant-height nc-AFM frequency shift image (CO tip) of an AO nanoribbon. **h** Corresponding chemical structure. Imaging parameters: (**a**) $I_T$ = 2 nA, $V_b$ = −0.50 V; **b** $I_T$ = 50 pA, $V_b$ = −1.00 V; (**c, g**) oscillation amplitude = 50 pm; (**e**) $I_T$ = 130 pA, $V_b$ = +1.09 V; (**f**) $I_T$ = 60 pA, $V_b$ = −0.50 V. All images acquired at $T$ = 4.3–7.0 K.

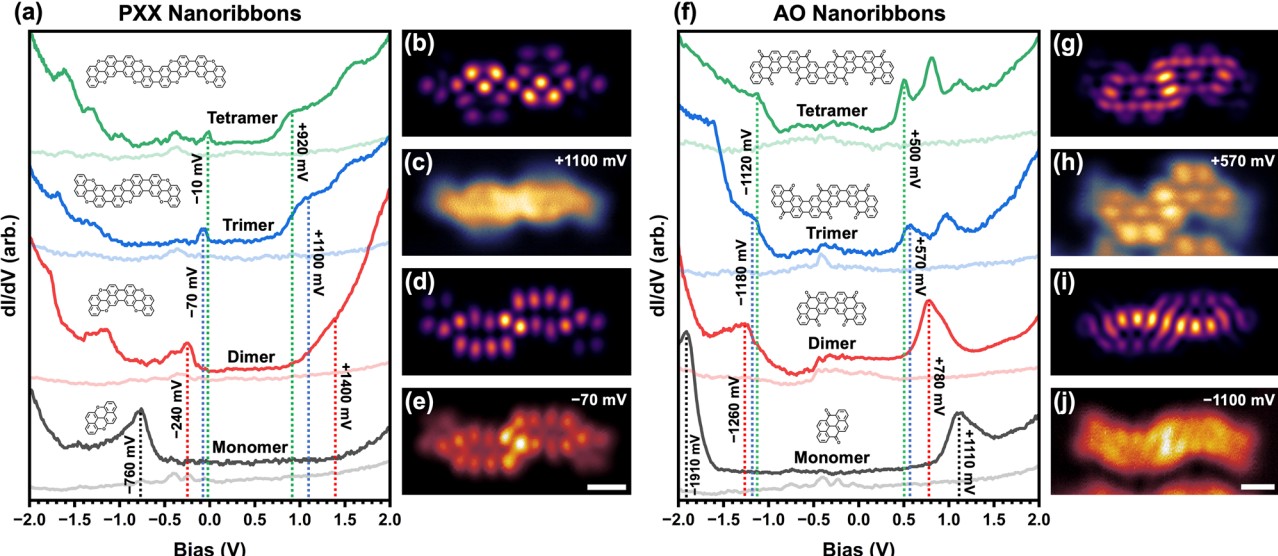

**Fig. 3 | Electronic structure of pure PXX and AO nanoribbons on Au(111). a, f** dI/dV spectra of PXX and AO nanoribbons of increasing lengths (monomers to tetramers), showing a progressive narrowing of the energy gap. Dotted lines mark the energetic positions of the frontier resonances. The faint lines for each nanoribbon are the Au(111) reference spectra. **c, e** Constant-height dI/dV images of a PXX trimer acquired at voltages corresponding to the lowest unoccupied and highest occupied resonances, respectively; **h, j** analogous images for an AO trimer (scale bars: 500 pm). **b, d, g, i** are simulated dI/dV images of the same resonances, using a probe with a 25% s- and 75% p-wave character to model the CO-functionalised tip. Measurement positions are indicated in Fig. S14.

resonances of PXX nanoribbons are significantly broader than the occupied states, a feature previously attributed to lifetime effects in surface-supported molecular systems[66].

Longer AO nanoribbons exhibit a pronounced acceptor character, with the lowest unoccupied state shifting to lower energies as the ribbon length increases. This resonance appears at +500 mV for the AO tetramer, significantly below that of the PXX tetramer (+920 mV), while the highest occupied state is over 1 eV deeper. Interestingly, the energy gap of the AO tetramer (1.62 eV) is substantially larger than that of its PXX counterparts (0.93 eV), despite both being predicted to have identical gas-phase gaps of 1.63 eV. This discrepancy likely arises from differences in substrate interaction, charge transfer and electrostatic screening. Decoupling the nanoribbons from the metallic substrate would likely reduce this difference.

Having established that the on-surface homopolymerisation of Br₂PXX and VO3 successfully yields donor and acceptor nanoribbons, respectively, we extended this strategy to mixed systems by co-depositing both precursors onto a Au(111) surface. Subsequent annealing to 473 K induced copolymerisation via the same surface-catalysed Ullmann coupling and dehydrogenation, resulting in large islands composed of various nanoribbons interspersed with dissociated Br atoms (Fig. 4a, b). High-resolution BR-STM imaging confirms that many nanoribbons incorporate both PXX and AO units, demonstrating the successful formation of D-A nanoribbons. The two components are readily distinguished in BR-STM and nc-AFM images due to their contrasting central ring features: the pyranopyranyl units of PXX appear darker, while the quinoidal moieties of AO show a brighter contrast with sharp features corresponding to their ketone groups[67]. Statistical analysis of the three possible junction types (D-D, D-A and A-A, see Fig. S18 and Table S1) reveals a clear preference for intermixing, with alternating D-A sequences occurring more frequently than D or A-block segregation. Interestingly, D-A junctions between PXX and AO units consistently result in straight ribbon segments, unlike D-D or A-A junctions, which often introduce bends (Fig. S19). This is likely due to the different positions of the bromine atoms in the two precursors. As a result, block-type ribbons tend to exhibit directional changes at the homojunctions, while alternating D-A sequences remain mostly straight.

While a complete analysis of the electronic structure of mixed D-A nanoribbons across different lengths and sequences lies beyond the scope of this study, we focus here on the characterisation of all four distinct PXX-AO trimers and the corresponding mixed dimer, using STS (Fig. 4c) and dI/dV imaging (Fig. 4d–h). This choice also reflects the expectation that these short oligomers capture the main changes in electronic structure, based on the trends previously observed for homopolymer nanoribbons. While the full set of STS data may appear complex at first glance, a clear initial trend can be identified: trimers containing two AO units (blue and purple spectra in Fig. 4c) exhibit lower HOMO and LUMO levels than those with a higher PXX content (red and green), as also shown in Fig. S23.

This trend can be effectively interpreted using a simplified linear combination of molecular orbitals (LCMO) approach, described in Section 21 of the SI. Within this framework, the HOMO and LUMO energies of a trimer are estimated from those of its constituent monomers and dimers, with variational corrections that scale inversely with the energy spacing between their corresponding orbitals. As a result, trimers with two A (or two D) units show stronger orbital mixing and larger energy shifts—downward for the LUMO and upward for the HOMO—reflecting enhanced acceptor (or donor) character. This simple method (Fig. S23c) qualitatively reproduces the experimental trends (Fig. S23a) and generally aligns well with more accurate DFT calculations (Fig. S23b), while offering a convenient tool for interpreting and predicting the electronic structures of mixed D-A oligomers.

The STS data further reveal that the two block-type trimers (PXX-PXX-AO and AO-AO-PXX) exhibit energy gaps 0.15–0.2 eV smaller than their alternating polymers, despite having the same monomer composition. This narrowing likely arises from enhanced orbital overlap between adjacent identical units, which promotes greater intrablock delocalisation and stabilises the frontier orbitals. While this effect is not captured by the simplified LCMO model (see Section 21), it is correctly reproduced by DFT (Table S2). However, some features are not fully accounted for by the DFT calculations either, most likely due to differences between the idealised vacuum environment assumed in the simulations and the real experimental conditions. In particular, the measured trimers are adsorbed on Au(111) and subject to local

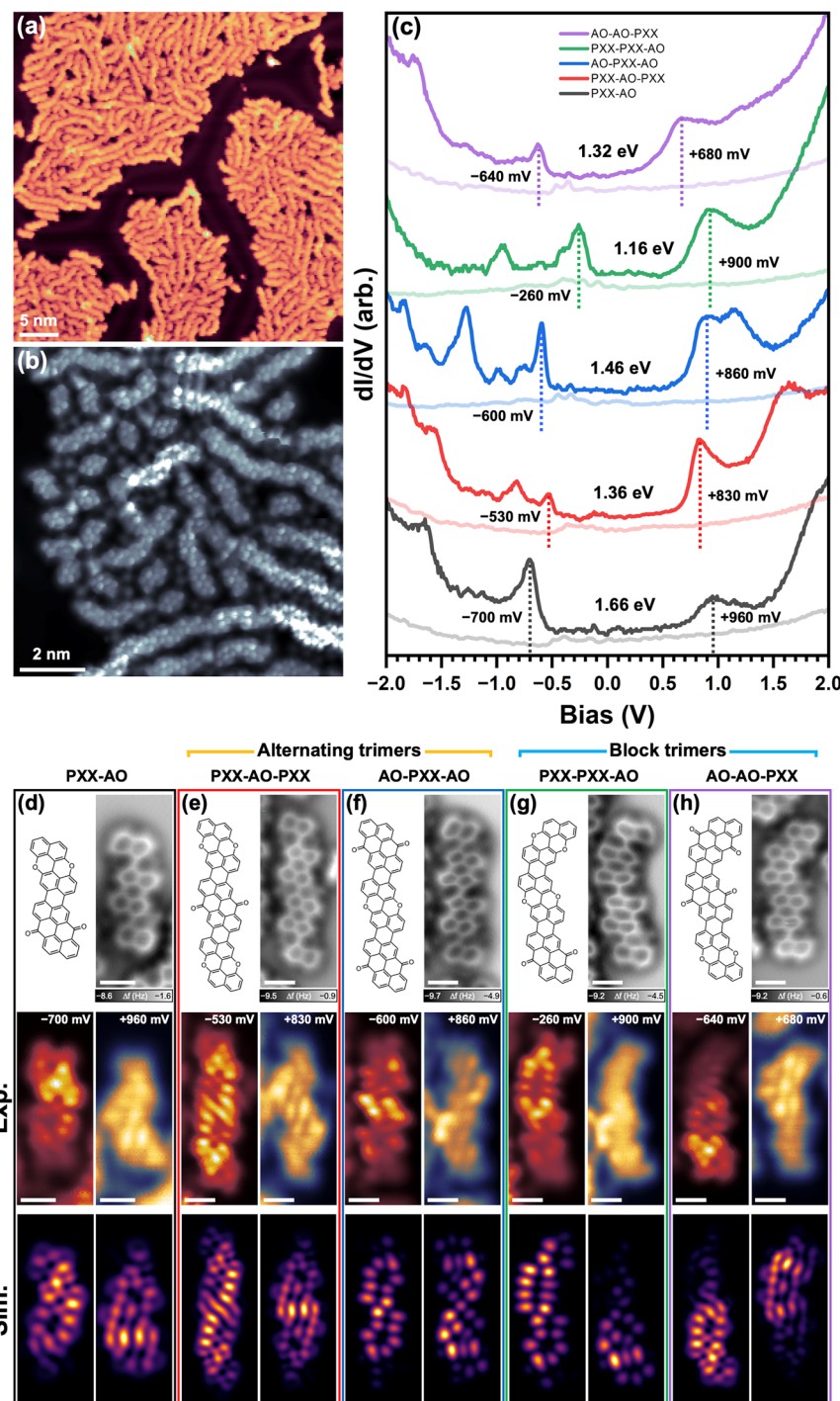

**Fig. 4 | Mixed PXX-AO nanoribbons on Au(111). a** Large-scale STM image showing mixed PXX-AO nanoribbons and interspersed Br atoms. **b** Smaller scale constant-height BR-STM image of the same sample. **c** dI/dV spectra of the PXX-AO dimer and of all possible mixed PXX-AO trimers. Frontier orbital resonances are marked by dashed vertical lines; energy gaps are indicated. The faint lines for each nanoribbon are the Au(111) reference spectra. **d–h** Chemical structures, nc-AFM images, constant height dI/dV maps (CO tip) and simulated dI/dV images of the PXX-AO dimer and each trimer. For (**d, e, h**), a probe with 25% s- and 75% p-wave character was used to model the CO-functionalised tip. For (**f, g**), a probe with 75% s- and 25% p-wave character was used instead. Scale bars: 500 pm. dI/dV measurement locations are shown in Fig. S20. Imaging parameters: **a** $I_T$ = 170 pA, $V_b$ = −0.89 V; **b** $V_b$ = +0.03 V; (**d–h**) nc-AFM oscillation amplitude = 50 pm.

variations in their surroundings, including possible charge transfer effects and the presence of co-adsorbed Br atoms (as shown by the 'pure' donor or acceptor nanoribbons, further annealing to fully remove Br leads to longer ribbons, complicating the preservation of short mixed oligomers). To assess the magnitude of these effects, we compared dI/dV spectra from several PXX-AO dimers with different surroundings and terminations. The resulting variations in

the positions of the frontier resonances—up to approximately 0.1 eV (Fig. S24)—are in line with those measured for different Br₂PXX precursors (Fig. S25).

Constant height dI/dV images recorded with a CO-functionalised tip (Fig. 4d–h) reveal the spatial distribution of the frontier states. The highest occupied states are primarily localised on the donor PXX units, while the lowest unoccupied states show greater intensity over the

acceptor AO units. A degree of orbital extension across the PXX-AO heterojunctions is expected due to the formation of a shared perylene moiety. This is indeed observed, with the HOMO states partially delocalised into adjacent AO units, for example, in the AO-AO-PXX trimer shown in Fig. 4h. Comparison with the corresponding simulated dI/dV images (displayed below the experimental dI/dV images in Fig. 4) shows excellent agreement in both shape and spatial distribution, considering the *p*-wave character of the CO tip. Variations due to tip effects are discussed in Sections 24 and 25 of the SI.

These results demonstrate the feasibility of synthesising D-A nanoribbons via on-surface (co)polymerisation by tuning the donor and acceptor character of the molecular building blocks used as (co) monomers, closely mirroring the design strategies employed in solution-phase conjugated polymer chemistry. Crucially, the atomic precision afforded by on-surface synthesis, combined with the advanced characterisation techniques of UHV surface science, enables structural and electronic analysis of the resulting nanoribbons with unmatched spatial resolution.

In this work, we have demonstrated the on-surface synthesis of ultra-narrow donor-acceptor nanoribbons via the surface-assisted polymerisation of PXX and AO molecular precursors and characterised their structure and electronic properties at submolecular resolution using BR-STM, nc-AFM and STS. Homopolymerisation of each precursor yields structurally well-defined nanoribbons with distinct electronic character: PXX-based ribbons exhibit a progressive increase in donor strength with length, while AO-based ribbons display pronounced electron-accepting behaviour. Copolymerisation of the two precursors produces mixed donor–acceptor nanoribbons, where the local electronic structure is strongly modulated by monomer sequence. The experimentally observed orbital distributions and energy levels are well described by gas-phase DFT calculations, while the overall trends in level shifts are captured by a simple linear combination of molecular orbitals model, offering an intuitive framework for predicting the electronic structure of D−A oligomers. This strategy parallels solution-phase D−A copolymer design yet harnesses the atomic precision and high-resolution characterisation uniquely enabled by on-surface synthesis in ultrahigh vacuum. Together, these results open new avenues for the atomically precise engineering of functional π-conjugated nanostructures with tuneable optoelectronic characteristics.

## Methods
### Materials
The synthesis of $Br_2PXX$ is described in our previous work[61] and reported in Section 27 of the SI. VO3 was purchased commercially from Carbosynth.

### STM and nc-AFM
STM, nc-AFM and STS experiments were performed on two systems: a Createc LT-STM at 7 K and a Scienta-Omicron LT-SPM at 4.3 K. The Au(111) crystal was prepared via standard cycles of sputtering and annealing. $Br_2PXX$ and VO3 were both deposited onto the room temperature surface via sublimation using an organic molecular beam deposition system (Dodecon Nanotechnology). PXX was sublimed at 165 °C, $Br_2PXX$ at 210 °C, and VO3 at 270 °C. Typical bias voltages ($V_b$) in the range of −2 V to +2 V and tunnelling current set points ($I_T$) of 50–300 pA were used for standard STM imaging.

To obtain bond-resolving nc-AFM and STM images, CO was deposited onto the cold (7–10 K) Au(111) surface by leaking CO gas into the chamber at a pressure of $10^{-7}$ mbar for 20–30 s. It was then picked up from the surface either by scanning with typical tunnelling parameters (negative bias voltage) or by approaching the tip over CO clusters. nc-AFM images were recorded with a QPlus sensor ($Q \approx 120k$) at a constant height with a typical oscillation amplitude of 50 pm. BR-STM images were recorded with a low bias voltage (5–40 mV) at a constant height with the tip in close proximity to the molecules.

### STS
dI/dV measurements were acquired using an internal lock-in with oscillation amplitudes of 5–40 mV and frequencies at 780–1080 Hz. Constant height dI/dV images were recorded using similar oscillation parameters. For some of the nanoribbons, the constant height dI/dV spectra for positive and negative bias voltages were recorded separately. This is due to spatial variations in the intensity of the dI/dV signal coming from the different occupied and unoccupied states. In some cases, dI/dV images were FFT-filtered to remove high-frequency electronic noise.

### XPS
XPS experiments were performed on an Omicron Multiprobe system with a monochromated Al Kα source and a photon energy of 1486.7 eV in an analysis chamber with a base pressure lower than $1 \times 10^{-10}$ mbar. The analyser work function and binding energy scale were calibrated according to photoemission peaks and the Fermi level position of a polycrystalline silver foil with a known work function. A pass energy of 10 eV (resolution approximately 0.47 eV) was used for recording core-level spectra from a 1.1 mm diameter area on the surface. Samples were prepared in an adjoining preparation chamber, with molecules deposited via sublimation using an organic molecular beam deposition system (Dodecon Nanotechnology), before being transferred to the analysis chamber for XPS measurements. Annealing was performed in the main analysis chamber using a filament built into the manipulator and temperature monitoring via a chromel/alumel thermocouple in close proximity to the sample.

### DFT
All DFT calculations were carried out using the mixed Gaussian and Plane-Waves (GPW) method implemented in the CP2K package[68]. Given the sensitivity of intermolecular interactions and electronic level alignment to the choice of exchange-correlation (XC) functional, we explored a range of functionals with differing levels of sophistication. Specifically, we employed the fully self-consistent non-local vdW-DF XC functional[69] as well as two different hybrid XC functionals: B3LYP[70] and HSE06[71].

All calculations were performed in the gas phase, with no periodic boundary conditions, starting from the experimental molecular geometries. The Kohn−Sham orbitals were expanded in a triple-zeta valence plus double polarization (TZV2P) Gaussian basis set. Core electrons were treated using Goedecker−Teter−Hutter (GTH) pseudopotentials[72], with four, one, six, five, and three valence electrons for carbon, hydrogen, oxygen, nitrogen, and boron atoms, respectively. The plane-wave cutoff for the finest level of the multigrid used to solve the Poisson equation[68] was set to 400 Ry. These settings were sufficient to converge the total energy to 4 meV/atom.

## Data availability
The data supporting this paper have been deposited in a publicly available Zenodo repository with DOI:10.5281/zenodo.18851738. The data in question includes: (a) the STM, BR-STM, STS, and nc-AFM measurements presented in this paper and; (b) the input files, geometries, and post-processing scripts utilised to perform and analyse the DFT simulations (and simulated STM images) featured in this work. All data are available from the corresponding author upon request.

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

## Acknowledgements

We thank Richard Walton for kindly providing the VO3 precursor used in this study. J. Lawrence acknowledges the support from the Agency for Science, Technology & Research (A*STAR), Singapore, through its MTC YIRG Grant (Project ID: M23M7c0121). G.C. acknowledges funding from the EU through the European Research Council (ERC) Grant "VISUALMS" (Project ID: 308115). G.C.S. gratefully acknowledges the use of SULIS, which was funded by the EPSRC (EP/T022108/1), via the HPC Midlands+ Consortium. G.C.S. and F.B. would also like to acknowledge the high-performance computing facilities provided by the Scientific Computing Research Technology Platform (SCRTP) at the University of Warwick. D.B. gratefully acknowledges the EU through the MSCA-RISE (project: VIT, No. 101008237) and the University of Vienna for generous financial support. J. Lu acknowledges the support of the Ministry of Education (MOE) Singapore (MOE-T2EP10124-0004).

## Author contributions

J. Lawrence performed the STM, BR-STM, STS, and nc-AFM measurements, carried out the XPS measurements, analysed the data, proposed the use of complementary donor and acceptor comonomers for the on-surface synthetic strategy, and co-wrote the first version of the manuscript. L.Đ. synthesised Br$_2$PXX. F.B. contributed to perform, analyse and interpret the DFT calculations. H.P. performed an initial set of DFT calculations. M.W. contributed to the XPS measurements and data analysis. J. Lu supervised the nc-AFM measurements. G.C.S. performed, analysed and interpreted the DFT calculations, generated the simulated dI/dV images, and supervised the computational aspects of the work. D.B. proposed the on-surface synthesis of PXX nanoribbons and supervised the synthetic work. G.C. supervised the STM, BR-STM, and STS measurements, developed the simplified LCMO model, co-wrote the first version of the manuscript, and supervised and coordinated the entire project. All authors contributed to the editing and revision of the manuscript.

## Competing interests

The authors declare no competing interests.
