## [Transparent Peer Review file · Nature Communications]

Ultra-narrow donor-acceptor nanoribbons

Corresponding Author: Professor Giovanni Costantini

Version 0:

Reviewer comments:

Reviewer #1

(Remarks to the Author)

This manuscript describes the on-surface synthesis of donor-acceptor nanoribbons by the co-polymerisation of brominated derivatives of peri-xanthenoxanthene and anthanthrone. The authors characterise these nanoribbons, and those synthesised by homo-polymerization of each precursor, by combining bond-resolved scanning tunnelling microscopy and spectroscopy (dI/dV) with non-contact atomic force microscopy. Linear combination of molecular orbitals and DFT calculations excluding the gold substrate are deployed to explain the shift in energy of the frontier orbitals detected in dI/dV spectra of donor-acceptor nanoribbons of variable length and composition and to understand the spatial distribution in dI/dV images.

The manuscript is well written and most of the specialised concepts announced in the manuscript are explained either in the text or in the supplementary information. I could not find any flaws in the method, interpretation, discussion, and conclusions that would prevent publication in Nature Communications. The results are relevant to the on-surface chemistry field as well as to the field of organic electronics. Consequently, I recommend the publication of the manuscript after minor revision of the text and the additional information.

Line 40: Maybe it is candidates instead of condidates.

Line 51: I suggest using the word "nanoribbons" instead of NRs (which is only defined when referring to graphene nanoribbons). There is no much difference between using the word and the acronym, but the fewer acronyms, the clearer the text.

Line 74: Please, define N.

Lines 161 to 175: In my version of the manuscript, the text in dose lines is a word-by-word copy of the text in lines 137 to 151.

Lines 180 and 186: In my opinion, the words "modelling" and "model" are misleading. "Structures" is more appropriate. In Figs. S10 and S12, the authors superimpose and assign tentative structures to the molecular arrangements. "Modelling" is normally associated with structures derived from first-principles calculations.

Line 193: It would be helpful to indicate how panels a, b, c in Fig.S13 relate to each other.

Line 219: The orientation of the structures in the 2nd and 4th panels from the left for the PXX nanoribbons seems to be mirrored with respect to the experimental image.

Calls to Fig.S6 in the text of Section 16 of the supplementary information should be Fig.S7 instead.

In Figs. 3 and S15, the dimers and trimers seem to be in a more isolated environment than the counterparts in Figs. 4 and S18. Have the authors tried to manipulate the donor-acceptor dimers and trimers out of the islands they assemble to ensure that proximity effects have no influence in the spectroscopic measurements?

Reviewer #2

(Remarks to the Author)

In this work, Lawrence and co-workers report the on-surface synthesis of donor-acceptor graphene-based nanoribbons. Their structural and electronic properties are characterized by scanning probe microscopy (SPM), complemented by gas-phase calculations based on density functional theory (DFT).

Although heteroatom doping in graphene nanoribbons (GNRs) has been reported, donor acceptor (D-A) GNR architectures remain largely unexplored; I agree with the authors on this point.

The choice of donor like (Br₂PXX) and acceptor like (Br₂AO) molecular precursors allows modulation of the electronic structure in ultranarrow D-A nanoribbons, with composition dependent frontier state distributions.

SPM and XPS provide clear evidence of successful GNR fabrication. DFT simulations with hybrid functionals aid the interpretation of STS data.

In my view, these results broaden the options for tuning the electronic properties of atomically precise carbon-based nanostructures and deserve publication in Nature Communications, provided the authors address the following minor comments.

1)
“The predominance of alternating junctions—causing the wavy morphology of most NRs (Figs. 2(c) and 2(g))—suggests that the initial relative orientation of the reacting C-Br groups plays a key role in determining coupling efficiency. Specifically, the reaction appears to be favoured when the two C-Br (and corresponding C-H) groups are aligned face-to-face. The same steric and configurational considerations apply to the VO₃ precursors during the formation of the acceptor NRs (see below).”

I would say that after debromination, radical carbons tend to couple to form C-C bonds. From Fig. S4, two pathways seem plausible:

for molecules oriented as in (b), initial coupling is followed by dehydrogenation, completing “one alternating block”;

for molecules oriented as in (a), single units may rotate about the first C–C bond before dehydrogenation, again yielding “one alternating block”.

Direct, concerted formation of two C-C bonds as suggested in (a) appears less probable.

These considerations are speculative and would require additional, demanding, DFT calculations for confirmation, but, maybe are a possible interpretation of the experimental evidence. This speculation is also compatible with:

“For instance, molecules brominated at the 1-position instead of the 3-position (1 in Fig. S5) tend to form ‘straight’ rather than the regular ‘alternating’ junctions, while tribrominated species (2 in Fig. S5) may result in branched structures.”

2)
“In many cases, bright junctions are observed connecting the termini of adjacent ribbons. These are likely the result of precursor misalignment during polymerisation, leading to the formation of a single C-C bond rather than a fully fused junction (Fig. S13).”

I fully agree. Because AO NRs attract each other, a defect in a partially reacted polymer could trigger a nearby defect in an adjacent partially reacted polymer, as suggested in Fig. S13(b).

3)
“Interestingly, the energy gap of the AO tetramer (1.62 eV) is substantially larger than that of its PXX counterparts (0.93 eV), despite both being predicted to have identical gas-phase gaps of 1.63 eV. This discrepancy likely arises from differences in substrate interaction, charge transfer and electrostatic screening. Decoupling the NRs from the metallic substrate would likely reduce this difference.”

Did the authors attempt to decouple the NRs from Au(111), as done in related studies? If so, what was the outcome? If not successful, what is the most likely reason?

4)
Table S2. GGA-DFT typically underestimates HOMO LUMO gaps. On Au(111), image-charge screening reduces experimental STS gaps, which can make GGA-DFT gaps appear coincidentally close to experiment. Gas-phase beyond-DFT estimates (e.g., GW) should yield larger gaps than STS on Au(111). Here, the HSE06 gaps are still smaller than the measured values; can the authors comment on this discrepancy?

5)
It would be valuable to compare DFT results (e.g., with CP2K) and STS for long ribbons forming a D-only||A-only heterojunction via sequential deposition of D precursors followed by A precursors. Did the authors attempt such heterojunctions, beyond the dimers and trimers discussed?

6)
"HOMO and LUMO are not physically meaningful, as Kohn-Sham orbitals are auxiliary constructs which absolute energies depend on both the choice of the exchange-correlation functional and the choice of the (arbitrary) energy zero."

How did the authors grant consistent energy alignment among different calculations?

Reviewer #3

(Remarks to the Author)

Please, find my comments for the authors in the attached file.

Version 1:

Reviewer comments:

Reviewer #2

(Remarks to the Author)

The authors have addressed all my comments and replied to my questions.

Following the constructive feedback from all reviewers, the authors have improved the manuscript, which in my opinion now deserves publication.

Reviewer #3

(Remarks to the Author)

The authors have thoroughly considered my initial concerns and improved their work. Most importantly, the identification of the HOMO and LUMO positions of the nanoribbons has been strengthened significantly through the addition of simulated STM images, which now supports the main results of the article to a satisfactory degree. The description of the LCAO model, discussion of the absolute energy values, interpretation of the XPS results, and further details were clarified. Overall, I appreciate the improvements made in the revision and do not have further comments.

Point-by-point response to the Reviewers' comments

We thank the Reviewers for their thorough and careful evaluation of our manuscript. We are grateful for their constructive feedback and suggestions, which have allowed us to clarify several points and further strengthen the presentation of our work. As a result, we believe that the revised manuscript represents a significantly improved and clearer version of the paper.

We appreciate the very positive assessments provided by Reviewers #1 and #2. Reviewer #1 notes that “the results are relevant to the on-surface chemistry field as well as to the field of organic electronics” and recommends publication of the manuscript after minor revision. Reviewer #2 similarly recognises the significance of the work, stating that “these results broaden the options for tuning the electronic properties of atomically precise carbon-based nanostructures and deserve publication in Nature Communications” following the clarification of minor points. We have addressed all comments raised by these Reviewers carefully and in full.

Reviewer #3 also recognises the value of translating donor–acceptor strategies from solution chemistry to on-surface synthesis and acknowledges both the quality and the breadth of the experimental data. However, the Reviewer also raises a number of concerns related primarily to the analysis, interpretation, and presentation of the results. As detailed in our responses below, we believe that the issues raised by Reviewer #3 largely arise from passages where the original wording lacked sufficient clarity, or from points that can be addressed through the adoption of the Reviewer’s constructive suggestions, many of which have been incorporated into the revised manuscript. We are therefore grateful to Reviewer #3 for their exceptionally thorough and detailed comments, which have helped us to improve clarity and precision throughout both the main text and the Supplementary Information (SI).

In the following, we respond to each comment raised by the Reviewers in detail. For Reviewer #3, we first address their general comments before responding individually to the specific points they made. All corresponding changes to the main text and the SI are highlighted in blue.

Reviewer #1 (Remarks to the Author):

Line 40: Maybe it is candidates instead of condidates.

We thank the Reviewer for pointing out this mistake which has been corrected.

Line 51: I suggest using the word "nanoribbons" instead of NRs (which is only defined when referring to graphene nanoribbons). There is no much difference between using the word and the acronym, but the fewer acronyms, the clearer the text.

We agree with this suggestion and have consequently replaced the acronym with the full term in all instances where it appeared, both in the main paper and in the SI.

Line 74: Please, define N.

We apologise for not having defined this parameter, which is sometimes used in the literature as a measure of the nanoribbon width. Following the Reviewer's comment, we have removed this parameter, as it is not particularly relevant and is not used elsewhere in either the main text or the SI.

Lines 161 to 175: In my version of the manuscript, the text in dose lines is a word-by-word copy of the text in lines 137 to 151.

We are very grateful to the Reviewer for pointing out this error and apologise for the oversight, which has now been corrected.

Lines 180 and 186: In my opinion, the words "modelling" and "model" are misleading. "Structures" is more appropriate. In Figs. S10 and S12, the authors superimpose and assign tentative structures to the molecular arrangements. "Modelling" is normally associated with structures derived from first-principles calculations.

We thank the Reviewer for this helpful observation. We agree that the terms "modelling" and "model" may inadvertently suggest that the structures were obtained through computational simulations. In our case, they refer to molecular overlays and geometric assignments based on STM data. To avoid confusion, we have replaced "modelling" with "structural assignment" in the main text (referring to Fig. S10) and "model" with "molecular arrangement" (referring to Fig. S12) in the revised manuscript. We have also altered the wording of the corresponding SI figure captions. We believe these changes clarify our intended meaning.

Line 193: It would be helpful to indicate how panels a, b, c in Fig.S13 relate to each other.

We thank the Reviewer for noting that this information was missing and apologise for the omission. Panel (b) shows a zoomed region of the area displayed in panel (a). Panel (c) is an nc-AFM image of the same type of junctions, but from a separate experiment. To clarify this, we have added a small white dashed box in panel (a) to indicate the region shown in panel (b) and revised the caption of Fig. S13 accordingly:

"The dashed white box in (a) indicates the position of the zoomed image in (b)."

"(c) nc-AFM image of another example of the same type of terminal junctions between AO nanoribbons, recorded in a separate experiment."

Line 219: The orientation of the structures in the 2nd and 4th panels from the left for the PXX nanoribbons seems to be mirrored with respect to the experimental image.

We thank the Reviewer for pointing this out and have revised Fig. S14 so that the orientation of the chemical structures better matches with the STM images.

Calls to Fig.S6 in the text of Section 16 of the supplementary information should be Fig.S7 instead.

We thank the Reviewer for pointing out this error and apologise for the oversight, which has now been corrected.

In Figs. 3 and S15, the dimers and trimers seem to be in a more isolated environment than the counterparts in Figs. 4 and S18. Have the authors tried to manipulate the donor-acceptor dimers and trimers out of the islands they assemble to ensure that proximity effects have no influence in the spectroscopic measurements?

We thank the Reviewer for their comment and question. We expect that the position of the resonances in the dI/dV spectra to be slightly shifted when the nanoribbons are in different local environments, particularly when they are placed within large islands containing “free” bromine atoms that chemisorb on the underlying Au(111) surface. We did not attempt to manipulate the mixed nanoribbons outside these islands, as most of them were situated in configurations that would have required extensive ‘reconstruction’.

However, the influence of locally adsorbed Br atoms has been experimentally investigated using a model system of Br_2PXX molecules (Fig. S25), as well as the corresponding dimers (Fig. S24). As shown there, an up-shift of the resonance of up to approximately +100 mV can be expected for molecules located within or at the edge of such mixed islands. The mixed nanoribbon trimers shown in Fig. 4 of the main text and Fig. S20 of the SI are all located at the edge or within an island containing bromine/other nanoribbons and have at least one Br atom in close proximity. Consequently, we expect that a comparison between these systems should remain valid. Smaller variations due to the presence of other nanoribbons cannot be excluded, as is currently mentioned in the main text.

Reviewer #2 (Remarks to the Author):

1) *“The predominance of alternating junctions—causing the wavy morphology of most NRs (Figs. 2(c) and 2(g))—suggests that the initial relative orientation of the reacting C-Br groups plays a key role in determining coupling efficiency. Specifically, the reaction appears to be favoured when the two C-Br (and corresponding C-H) groups are aligned face-to-face. The same steric and configurational considerations apply to the VO₃ precursors during the formation of the acceptor NRs (see below).”*

I would say that after debromination, radical carbons tend to couple to form C-C bonds. From Fig. S4, two pathways seem plausible:

- *for molecules oriented as in (b), initial coupling is followed by dehydrogenation, completing “one alternating block”;*
- *for molecules oriented as in (a), single units may rotate about the first C-C bond before dehydrogenation, again yielding “one alternating block”.*

Direct, concerted formation of two C-C bonds as suggested in (a) appears less probable.

These considerations are speculative and would require additional, demanding, DFT calculations for confirmation, but, maybe are a possible interpretation of the experimental evidence. This speculation is also compatible with: “For instance, molecules brominated at the 1-position instead of the 3-position (1 in Fig. S5) tend to form ‘straight’ rather than the regular ‘alternating’ junctions, while tribrominated species (2 in Fig. S5) may result in branched structures.”

We thank the Reviewer for their insightful comments. We generally agree with their interpretation, with the caveat that, for the scenario shown in panel (a), rotation around the newly formed bond may be disfavoured depending on the nanoribbon growth mechanism. If the junction arises from the coupling of two pre-formed nanoribbons (similar to the step-growth in solution polymer chemistry), such a rotation would require one nanoribbon to flip in an energetically unfavourable manner after covalent bond formation. However, if growth proceeds via the reaction of individual precursor monomers with nanoribbon termini (similar to chain-growth in solution polymer chemistry), this scenario becomes more plausible, as it would only require reorientation of the terminal unit. These possibilities are discussed in Sections 4, 6 and 9 of the SI.

Connections between mismatched nanoribbon termini were observed much more frequently for the VO₃ (ketone) precursor (Fig. S13), which may indicate differences in the underlying growth mechanisms of the two precursors. Unfortunately, we do not have calculations to explicitly examine these mechanisms. As suggested by the Reviewer, such calculations would be computationally demanding, and we consider them to be beyond the scope of the present study.

To clarify our interpretation, we have revised the main text and added an explicit reference to the SI. The revised text now reads:

“The predominance of alternating junctions—causing the wavy morphology of most nanoribbons (Figs. 2(c) and 2(g))—suggests that the initial relative positions of the reacting C-Br groups play a key role in determining the coupling efficiency. Specifically, the reaction appears to be favoured when the radicals generated by precursor debromination are aligned face-to-face. The same steric and configurational considerations apply to the VO₃ precursors during the formation of the acceptor nanoribbons (see below). Further possible reaction mechanisms are discussed in Sections 4, 6 and 9 of the SI.”

2) “In many cases, bright junctions are observed connecting the termini of adjacent ribbons. These are likely the result of precursor misalignment during polymerisation, leading to the formation of a single C-C bond rather than a fully fused junction (Fig. S13).”

I fully agree. Because AO NRs attract each other, a defect in a partially reacted polymer could trigger a nearby defect in an adjacent partially reacted polymer, as suggested in Fig. S13(b).

We thank the Reviewer for this interesting suggestion. It is difficult to determine whether the self-assembly of the AO nanoribbons influences the reaction pathway, as their behaviour at the significantly higher reaction temperature (573 K) is not known. As noted in our previous response, this may be related to a different reaction mechanism or to distinct nanoribbon behaviours at the elevated temperatures required for growth. It is also possible that misaligned nanoribbon junctions simply preferentially self-assemble in this configuration upon cooling to lower temperatures, without being directly linked to the reaction itself.

3) “Interestingly, the energy gap of the AO tetramer (1.62 eV) is substantially larger than that of its PXX counterparts (0.93 eV), despite both being predicted to have identical gas-phase gaps of 1.63 eV. This discrepancy likely arises from differences in substrate interaction, charge transfer and electrostatic screening. Decoupling the NRs from the metallic substrate would likely reduce this difference.”

Did the authors attempt to decouple the NRs from Au(111), as done in related studies? If so, what was the outcome? If not successful, what is the most likely reason?

In the manuscript, we suggest that decoupling the nanoribbons from the metallic substrate (through one of several strategies reported in the literature) would likely reduce the difference between the energy gaps of the donor and acceptor monomers, bringing them closer to their predicted gas-phase values. However, we did not perform such experiments in the present study. We agree that this would be an interesting direction for future work, but believe it lies beyond the scope of the current manuscript.

4) Table S2. GGA-DFT typically underestimates HOMO LUMO gaps. On Au(111), image-charge screening reduces experimental STS gaps, which can make GGA-DFT gaps appear coincidentally close to experiment. Gas-phase beyond-DFT estimates (e.g., GW) should yield larger gaps than STS on Au(111). Here, the HSE06 gaps are still smaller than the measured values; can the authors comment on this discrepancy?

HSE06 is a screened, range-separated hybrid exchange–correlation functional. Extensive benchmark studies have shown that, despite its improved performance compared to semilocal functionals, HSE06 still tends to systematically underestimate experimental band gaps, typically by several tenths of an eV [e.g. Borlido *et al.*, Large-Scale Benchmark of Exchange–Correlation Functionals for the Determination of Electronic Band Gaps of Solids, *J. Chem. Theory Comput.* **15**, 5069–5079 (2019)]. Image-charge screening by the Au(111) substrate is expected to reduce the experimental STS gap; however, for the donor–acceptor nanoribbons studied here, this reduction does not appear to be sufficient to compensate for the intrinsic underestimation associated with HSE06. As a result, HSE06 band gaps smaller than those measured by STS are not unexpected. We agree with the Reviewer that a proper gas-phase GW calculation would be expected to yield larger gaps than both approaches; however, we also believe that such calculations are beyond the scope of the present work.

5) It would be valuable to compare DFT results (e.g., with CP2K) and STS for long ribbons forming a D-only||A-only heterojunction via sequential deposition of D precursors followed by A

precursors. Did the authors attempt such heterojunctions, beyond the dimers and trimers discussed?

We thank the Reviewer for this insightful suggestion. We agree that experiments on mixed nanoribbons comprising extended donor and acceptor segments would indeed be very interesting. Upon examining our existing data, we did not find much evidence of such nanoribbons forming spontaneously during the polymerisation of co-deposited donor and acceptor units; the probability of such sequences occurring randomly is in fact rather low. Therefore, the preparation of these systems would require an *ad hoc* synthesis, as the Reviewer suggests. While we agree that this would be an interesting direction for future work, we also believe that it lies beyond the scope of the present manuscript.

6) *“HOMO and LUMO are not physically meaningful, as Kohn-Sham orbitals are auxiliary constructs which absolute energies depend on both the choice of the exchange-correlation functional and the choice of the (arbitrary) energy zero.”*

How did the authors grant consistent energy alignment among different calculations?

We do not use absolute Kohn–Sham eigenvalues to interpret our results. All HOMO and LUMO energy levels were computed using an identical gas-phase setup (CP2K/GPW, TZV2P, GTH; see the Methods section), and the reported HOMO–LUMO gaps and level offsets are therefore compared only on a relative basis within the same exchange–correlation (XC) functional. Absolute energies are not used. Because all DFT calculations were performed in the gas phase using the same code, basis set, pseudopotentials and XC functional, the resulting trends in level ordering and energy gaps are internally consistent across the different systems.

Importantly, the experimental STS energies are reported relative to the Au(111) Fermi level instead. Our comparison between theory and experiment therefore focuses on (1) trends and relative shifts of the frontier states and their spatial character, and (2), where relevant, comparisons between HSE06 and other XC functionals to demonstrate the robustness of these trends (Table S3). We also explicitly acknowledge that environmental effects associated with Au(111), including screening, charge transfer and the presence of Br atoms, account for the remaining quantitative differences between gas-phase DFT and STS.

Reviewer #3 (Remarks to the Author):

Main comments

i. *“a reliable experimental identification of HOMO and LUMO of all investigated species, which is crucial for all discussion of the measured gaps, is currently lacking”*

This limitation has been addressed in the revised version of the manuscript by following the Reviewer’s helpful suggestion to use DFT-based simulated dI/dV images as the primary basis for comparison with the experimental data, rather than relying on molecular orbitals alone. We are grateful to the Reviewer for this suggestion, which, as they anticipated, enables a clearer and more unambiguous comparison with the experimental measurements and has strengthened the manuscript by allowing a more reliable identification of the frontier molecular orbitals.

ii. *“a significant mathematical issue is present in the simplified LCMO model and the conclusions drawn from it”*

We thank the Reviewer for raising this concern and for drawing attention to what may, at first sight, appear as a mathematical inconsistency in the simplified LCMO model. As clarified in our detailed response to comment 10 (see below), the underlying issue does not stem from an error in the model itself, but rather from our original unnecessarily intricate formulation of the resonance integrals and their scaling with the energy spacing, which made the description less transparent than intended.

Prompted by the Reviewer’s careful analysis, we have simplified the model and clarified the presentation in the SI by adopting fixed, physically reasonable values of the resonance integrals for different energy-spacing regimes. This removes the apparent ambiguity while leaving the qualitative behaviour and conclusions of the model unchanged. We apologise for this unnecessary complication in the original presentation and are grateful to the Reviewer for prompting a clearer and more transparent formulation.

Importantly, the simplified LCMO model remains an intentionally minimal, qualitative framework designed to capture robust trends in frontier orbital energies as a function of monomer sequence, rather than to provide a quantitatively predictive description. With the revised formulation, we believe this intent and the validity of the conclusions drawn from the model are now more clearly conveyed.

iii. *“specific details of the synthesis as investigated with STM and XPS remain unclear”*

We agree with the Reviewer that the detailed minutiae of the nanoribbon synthesis pathway are intrinsically complex and would require a dedicated, specialist study focused on reaction mechanisms and possibly kinetics to be resolved in full detail. However, this type of investigation is not the focus of the present work. By contrast, the main features of the synthesis that are relevant here are clearly and unequivocally established by our analysis and are consistently supported by the combined STM, bond-resolved SPM, and XPS data.

As addressed in detail in our responses to Comments 18–32, the atomic structure and chemical identity of the reaction products are unambiguously determined by bond-resolved STM and nc-AFM, while XPS provides complementary, supporting information on the thermal evolution of

the system, such as debromination and changes in the carbon bonding environment. These elements are sufficient for, and directly underpin, all conclusions drawn in this work.

Further mechanistic aspects of the surface reaction—such as the precise nature of transient intermediates or subtle changes during annealing—do not affect the identification, structure, or electronic properties of the nanoribbons investigated here. While such aspects may be of interest in studies specifically centred on reaction mechanisms, they lie beyond the scope of the present paper.

iv. *“Generally, the current manuscript is challenging to read as many pieces of information relevant for following a small part of the main text are distributed throughout the extensive SI without a clear order to help the reader.”*

We respectfully disagree with this comment. In the main manuscript, each section of the SI is clearly and unambiguously referenced at the relevant points in the text, ensuring that the reader can easily locate the corresponding material. Moreover, to facilitate navigation and allow the SI to be read as a standalone document, we have provided a comprehensive and detailed table of contents at its beginning.

v. *“This opens up questions about the focus of the paper. Is it the detailed understanding of the synthesis including a discussion of the possible (by-)products and coupling mechanisms or the detailed analysis and theoretical understanding of the electronic structure as suggested by the nice introduction? Whereas I find the synthesis in itself highly interesting as well and agree that it is a necessary prerequisite for all further studies of the products, the extensive investigation the authors present regarding both topics does not appear to fit in a concise way into this one communication.”*

We thank the Reviewer for this comment. The focus of the manuscript is twofold: first, to present the first successful on-surface synthesis of ultra-narrow donor–acceptor nanoribbons; and second, to characterise and rationalise their electronic properties. These central results are complemented by a detailed structural analysis of both the ideal nanoribbons and the by-products formed during the synthesis. As much of this information is essential for completeness yet ancillary to the main findings, we have included it in the SI. We agree with the Reviewer that providing such data is important for placing the synthesis in its proper experimental context and for enabling reproducibility in future studies, which we view as good scientific practice in line with conventions in synthetic research. We therefore believe that the level of detail provided strengthens, rather than detracts from, the clarity and value of the work. Moreover, the manuscript fully complies with the journal’s length requirements.

Detailed questions and comments

Identification of orbitals in STS and dI/dV images

1. *In Figure 3, Figure 4, Figure S15, and Figure S26 the authors compare dI/dV images taken at the specific voltages where orbitals, in most cases HOMO and LUMO, are expected from the STS and compare them to visualizations of DFT orbitals in order to confirm the energetic position of said orbitals (e.g. lines 209-212). Generally, this comparison is challenging to do directly by eye, especially due to the *p*-wave contribution which the authors mention and the fact that the CO-tip can lead to even more complex contrasts due to an overlap of both *p* and *s* contributions. Which one dominates depends for example on the tip-sample distance. A significantly more convincing agreement could be achieved easily by showing a direct comparison between the measured dI/dV images and simulated STM images, which is also the common approach in the literature. As the authors have already calculated the DFT orbitals, such simulations can be realized with little computational effort. A further advantage is that such simulations can be done assuming both an *s*- or *p*-wave tip and each measured image might be reproduced well by combing a specific ratio of *s* and *p* contributions in the simulation.*

We are grateful to the Reviewer for this suggestion and agree that simulated dI/dV images provide a more appropriate basis for comparison with the experimental data than molecular orbitals. Accordingly, we have performed these simulations, replaced the molecular orbitals in Figs. 3 and 4, and updated the corresponding references in the main text and figure captions. As anticipated by the Reviewer, the resulting simulated dI/dV images enable a more direct and unambiguous comparison with the experimental data and, as a consequence, allow for a more reliable assignment of the frontier molecular orbitals.

It is important to note that different tips were used throughout this work. We find that the images shown in Fig. 3 are most consistent with a predominantly *p*-wave-type tip, and we therefore used a ratio of 25% *s*- to 75% *p*-wave character in the simulations. In Fig. 4, two different tips with substantially different characteristics were used, as already discussed in Fig. S26. Tip 1 (Figs. 4(f) and 4(g)) exhibited a much stronger *s*-wave contribution, together with more distorted nc-AFM images; for this tip, we used a ratio of 75% *s* to 25% *p*. Tip 2 (Figs. 4(d), 4(e) and 4(h)) showed a markedly more *p*-wave-like character in the dI/dV images and less distorted nc-AFM contrast, and was therefore simulated using a 25% *s* to 75% *p* ratio.

Overall, we obtain very good agreement between the simulated and experimental images. Minor discrepancies may arise from the precise *s/p* mixing ratio chosen, as well as from known differences in how occupied and unoccupied states are imaged in the tunnelling process. Nevertheless, we consider the level of agreement to be fully adequate for the intended comparison. The caption of Fig. 4 and the text of Section 24 of the SI have been revised to include these details.

2. *In line with the previous point, I have to disagree in particular with the statement: “shows excellent agreement in both shape and spatial distribution, considering the *p*-wave character of the CO tip” in line 311 of the main text. Looking e.g. at the dI/dV images of the HOMO in Figure 4g and the LUMO in Figure 4h, I cannot recognize any *p*-wave contribution. The *p*-wave character should lead to a different number and different positions of nodes in the STM contrast compared to the appearance of an *s* wave image and thus the appearance of the DFT orbital. This can be understood as *p* wave images can be modeled employing the lateral derivative of the wave function (Gross et al., PRL 107, 086101 (2011), doi.org/10.1103/PhysRevLett.107.086101) as opposed to the wave*

function itself as is done for s-wave simulations. Additionally, the comparison between a metallic and a CO tip provided in Figure S15 b along with the description “generally similar for both tips, with finer detail resolved by the CO tip.” might rather suggest that the CO tip did not lead to exclusively p-wave tunneling here, but this can only be judged clearly by showing STM simulations.

We agree with the Reviewer’s comments; these points have been addressed in our response to the previous comment.

3. *In the STS presented in Figure 3a, how can one recognize the marked position of the LUMO for the PXX dimer? There appears to not be an obvious peak there.*

The LUMO-related peak for the PXX dimer manifests itself as a weak shoulder in the dI/dV spectra, which can be seen upon close inspection.

4. *The LUMO for the PXX monomer appears to be outside the measured bias voltage range in Figure 3a and no value is given in Table S2. As changes of the HOMO and LUMO positions with length and with composition of the ribbons are the main topic of the manuscript, this would be a crucial point of comparison and should be measured and discussed in comparison to the theory.*

We agree with the Reviewer that it would be ideal to clearly resolve this unoccupied resonance for PXX in the spectra. Unfortunately, attempts to acquire spectra at higher positive bias voltages did not reveal a distinct peak, but instead resulted in a monotonically increasing dI/dV signal. Probing at even higher voltages in an effort to access this resonance led to irreversible damage to the molecule. Consequently, we are unable to provide a reliable experimental determination of the energy of the first unoccupied resonance.

5. *In Figure 4c, the spectrum for PXX-AO exhibits some smaller peaks around -0.1 eV and +0.1 eV. What is their origin? Similarly, the orange curve in Figure S25 shows an additional peak at -250 mV. Generally, for all spectra given in the main text (Figure 3a,f, and Figure 4c), a spectrum of the Au(111) surface for comparison might be useful.*

We thank the Reviewer for this comment. The additional features observed in the spectra originate from tip-related states rather than molecular resonances. The visibility of these features varies depending on the molecule under investigation and on the local density of states of the underlying sample, which can enhance or suppress their appearance.

We also thank the Reviewer for the suggestion to include a reference spectrum of the Au(111) surface which we agree would be very useful for clarifying this point, and acknowledge that its absence was an omission. We have therefore added Au(111) reference spectra to Figs. 3 and 4 for direct comparison.

6. *In Figure S16 the broad HOMO peak appears clearly visible for the 2-mer and 3-mer. However, from the 4-mer on a significantly sharper peak appears (even sharper in b due to the chosen oscillation). (This is surprising in comparison to the statement “the unoccupied resonances of PXX NRs are significantly broader than the occupied states” in lines 229-230.) The sharp peak then shifts further to higher energies with NR length but significantly more slowly than the initial shift of the HOMO peak from the 2-mer to the 3-mer. As most clearly visible for the 4-mer for both tips, the broader HOMO peak might still be present underneath the sharper one, see around 0.06 V in Figure a and around -0.05 V*

in Figure b. Therefore, the authors should discuss other possible origins for the sharp peak close to the Fermi energy and its shape. For example, acenes (which are basically ultrathin zigzag edge NRs) are predicted to have open shell character increasing with their length, which leads to Kondo features close to the Fermi energy that have been measured with STS (Ruan et al., *J. Am. Chem. Soc.* 2025, 147, 4862–4870, doi.org/10.1021/jacs.4c13296) and can be fitted by appropriate functions (Fano or Frota line shapes).

We thank the Reviewer for this detailed observation. We agree that there may indeed be additional physics involved at the transition or discharging nanoribbon length (around six units), which could lead to the sharper appearance of the resonance as it approaches and crosses 0 V. However, as shown in Figs. S16 and S17, this peak clearly continues to shift through 0 V and is located at approximately +40 mV for a nanoribbon length of 21 units. This behaviour would not suggest a Kondo feature at that length.

We believe that a more detailed investigation of the nanoribbons at intermediate lengths could be extremely interesting and worthwhile to pursue in future work. However, such a study is beyond the scope of the present manuscript, which, as the Reviewer has noted, is already extensive and addresses multiple aspects of nanoribbon growth and properties.

To acknowledge this point explicitly, we have added the following sentence to Section 16 of the SI:

“It should also be noted that as the highest occupied resonance approaches and passes through 0 V, there may be other effects that contribute to its sharp appearance in the dI/dV spectra. In particular, there may be a transition regime in which the resonance becomes Kondo-like due to partial charge transfer to the surface. However, for longer ribbons the resonance clearly shifts beyond this regime, reaching approximately +40 mV for the nanoribbon shown in Figure S17. Whilst this transition regime could be interesting to examine in more detail, it lies beyond the scope of the present study and does not affect its general findings.”

7. *From Figures S14 and S20 (showing in the standard STM contrast, which easily appears wider than the geometry of the ribbon itself) and Figure S24 where the points are drawn next to the resolved bonds, it appears that all STS spectra presented are measured at the edges of the ribbons. I appreciate that the filled and empty states might be measured at separate positions depending on the spatial distributions of HOMO and LUMO, but the positions all appear deliberately quite far on the outside of the ribbons, which should be explained. Generally, comparing STS at different positions in one and the same ribbon might be useful for a more complete picture, especially in the center compared to the edges and at the different sections of the mixed ribbons. Specifically, at the edges complex and interesting electronic features like edge states might be present (Wang et al., *Nat Commun* 7, 11507 (2016). doi.org/10.1038/ncomms11507).*

This is a typical aspect of STS measurements on molecules adsorbed on metal surfaces, where the strongest signal from molecular resonances is most commonly obtained at the molecular edges. For this reason, all spectra presented here were recorded at the ribbon edges. This behaviour is generally attributed to the enhanced wavefunction “spill-out” at the edges, as well as to the reduced influence of nodal structures that can otherwise suppress tunnelling in the ribbon interior.

We have added a brief statement to Section 14 of the SI to clarify this point:

“It should also be noted that dI/dV spectra were generally recorded at the edges of the molecules/nanoribbons, as this is where the signal for the molecular resonances was found to be the strongest.”

8. *In Figure S14, it could aid the reader if the orientation of each structure was chosen the same as in the STM image (e.g. the structures in the top row second from the left and on the right would need to be mirrored).*

We thank the Reviewer for pointing this out and have revised Fig. S14 so that the orientation of the chemical structures better matches with the STM images.

9. *Line 274-276: HOMO and LUMO levels are described as “downshifted” and “upshifted”: With respect to the initial position of which ribbon/monomer is this shift?*

We thank the Reviewer for pointing out that the phrasing of this sentence was unclear. We have accordingly revised it to read:

“trimers containing two AO units (blue and purple spectra in Fig. 4(c)) exhibit lower HOMO and LUMO levels than those with a higher PXX content (red and green), as also shown in Fig. S23.”

LCMO model and comparison of theoretical to measured band gaps

10. *In lines 280-281 of the main text the authors claim that the LCMO model contains “variational corrections that scale inversely with the energy spacing”. Similar and more detailed descriptions are found in the SI, e.g. “This is because of the large values of Δ_{HOMO} and Δ_{LUMO} in equations (5) and (6), resulting in rather small corrections to $_{\text{HOMO}}D$ and $_{\text{LUMO}}A$, respectively.” and “because of the small value of Δ_{HOMO} at the denominator in equation (5).” I need to disagree with these statements and the conclusions drawn from them in the main text and the SI (pages 20 to 22). Whereas in Equations 5 and 6 it does appear that the correction factor decreases with the Δ as it appears in the denominator, the assumption that β is proportional to Δ leads to a Δ^2 in the numerator, thus leaving a Δ in the numerator (instead of the denominator) after reducing the fraction. Thus, the correction factor mathematically speaking actually increases with Δ . This appears to be compensated by choosing a smaller proportionality factor (the 2, 0.5, or 0.2 at the end of page 19 of the SI) within the β when the Δ is larger, such that in total a smaller correction factor still results for larger Δ . However, the origin of the proportionality factor and why it should be smaller for larger Δ remains unclear. The authors give the information that these factors appear appropriate “a posteriori” because the results agree with those from DFT. Overall, the model thus appears more like a fitting procedure with the proportionality factors being the resulting fit parameters that describe the measured trends rather than being a theoretical model from which conclusions could be drawn in comparison to DFT results as is currently claimed.*

We are very thankful to the Reviewer for this insightful and remarkably detailed comment. By meticulously examining our description of how the variational corrections scale with the energy spacing, they prompted us to revisit certain details of our simplified LCMO model, streamline the calculations of these corrections and clarify the explanation in the SI.

The Reviewer is indeed correct in pointing out that, in order to evaluate how the variational corrections in Equations (5) and (6) scale with the energy spacing ΔE , one must also consider how the resonance integrals β depend on ΔE . It is further correct that, according to our original description, β was assumed to scale proportionally with ΔE , which would in principle imply that the variational corrections are proportional to ΔE , since β appears squared in Equations (5) and (6). However, as also correctly noted by the Reviewer, in our original formulation the proportionality factor linking β to ΔE was itself dependent on ΔE , resulting in an overall inverse dependence of the variational correction on ΔE . Consequently, the statements in the manuscript and the SI (for example, the phrase “variational corrections that scale inversely with the energy spacing” cited by the Reviewer) are actually correct.

Nonetheless, this very perceptive observation by the Reviewer made us realise that our initial choice of approximate values for the resonance integrals β was unnecessarily intricate. In response, we have now adopted a simpler formulation in which β is taken as 0.2 eV for larger energy spacings ($\Delta E > 1$ eV) and 0.08 eV for smaller energy spacings ($\Delta E < 1$ eV). It should be noted that this choice still results in an overall inverse scaling of the variational corrections with the energy spacing. As in the previous version of the model, we also note that the choice of these specific values is justified a posteriori by the good agreement between the approximated LCMO energy values and those obtained from DFT calculations. Different β values lead to variations in the absolute HOMO and LUMO energy levels but, importantly, the overall trends remain unchanged.

Accordingly, we have updated the text in the SI to reflect this new choice of β values and have revised the corresponding HOMO and LUMO energy levels for the donor and acceptor oligomers (in Table S2, Fig. S21, S22 and S23). While these numerical values differ slightly (by few percents) from those reported in the previous version, this is not a concern, as the model is solely aimed at capturing qualitative trends, which remain unchanged.

Finally, we respectfully disagree with the Reviewer’s interpretation that our approach represents a fitting procedure rather than a very simple theoretical model. As explicitly stated in the SI, our goal was not to develop a full LCMO model, which would require detailed knowledge of the molecular orbital wave functions of the donor and acceptor monomers and the explicit form of the system Hamiltonian. Instead, our simplified approach aims to capture the qualitative trends in the frontier orbital energies of donor–acceptor oligomers as a function of their monomer sequence. The assumptions made regarding the values of β are deliberately simple, yet they successfully reproduce the main trends observed across all examined oligomers. Importantly, these trends remain robust even when the β values are varied within reasonable limits. This point has been explicitly reinstated and discussed in the revised version of the final part of Section 21 of the SI. We therefore believe that it is appropriate to describe this as an extremely simple model that effectively captures the key electronic trends and provides an intuitive framework for understanding their underlying behaviour.

11. In Table S2, the authors report absolute values for HOMO and LUMO energies as well as resulting gaps comparing the measurement and the DFT as well as the LCMO model. The absolute energetic positions of the measured HOMO and LUMO largely deviate from the results of DFT (and the LCMO model), which appears to not be clearly mentioned or discussed anywhere in the text. Whereas it is well-known that the absolute energies from DFT often vary from experimental ones, deviations between 4 and 5 eV seem rather significant. For example, the role of the surface (which is mentioned by the authors in line 294 in the context of the gaps) should be discussed in the context of the absolute energy levels here as well. Literature examples of such large energy level changes due to adsorption on a surface might support the idea or the authors could perform a DFT

calculation including the surface on just one short example ribbon to test this. Due to these large deviations, a reliable identification of the specific measured orbitals by means of simulated STM images (as mentioned above) is even more crucial.

We thank the Reviewer for raising this point and for highlighting what may, at first sight, appear to be a substantial discrepancy (4–5 eV) between the experimental and calculated absolute HOMO and LUMO energies reported in Table S2. However, this is only an apparent discrepancy, which primarily arises from the fact that the two datasets are referenced to different energy zeroes. The STS values are measured relative to the Fermi level of the Au(111) substrate, whereas the DFT (HSE06) and LCMO results correspond to isolated oligomers in the gas phase and are therefore naturally referenced to the vacuum level. As a consequence, the absolute values differ approximately by the work function of Au(111), which is about 5.2–5.3 eV, in addition to any interface-induced renormalisation effects.

When the experimental STS energies are shifted by a typical Au(111) work function, the discrepancy with gas-phase HSE06 is reduced to approximately 0.5–1.0 eV for the individual HOMO and LUMO levels of the oligomers listed in Table S2. For example, the AO monomer (A) exhibits STS resonances at –1.91 V (HOMO) and +1.11 V (LUMO) relative to the Fermi level, while the corresponding gas-phase HSE06 values are –6.04 eV and –3.78 eV relative to vacuum. After accounting for the Au(111) work function, the experimental levels correspond to approximately –7.2 eV and –4.2 eV on the vacuum scale, indicating stabilisation of about 1.2 eV and 0.4 eV, respectively, compared to the gas-phase HSE06 energies. Comparable shifts are obtained for the other oligomers in Table S2.

This degree of stabilisation is entirely consistent with expectations for molecules adsorbed on a polarizable metal substrate. Image-charge screening, interface dipoles and possible charge transfer are well known to lower the energies of molecular cationic and anionic states and to reduce frontier gaps by roughly 1–2 eV for conjugated molecules on metal or graphite surfaces. We therefore do not expect gas-phase HSE06 calculations, nor the simplified LCMO model built upon them, to reproduce the absolute level alignment on Au(111). Instead, these approaches are used to capture intrinsic trends in frontier orbital energies as a function of oligomer length and donor–acceptor composition.

In order to clarify this point, we have added two paragraphs to Section 1 and Section 21 of the SI, which respectively state:

“As these calculations were performed for isolated oligomers in the gas phase, the natural and consistent choice here was to reference the energy levels to the vacuum level.”

and

“Finally, it should be noted that the apparently large discrepancy (4–5 eV) between the experimental and calculated absolute HOMO and LUMO energies reported in Table S2 arises from the fact that the two datasets are referenced to different energy zeroes. The STS values are measured relative to the Fermi level of the Au(111) substrate, whereas the DFT (HSE06) and LCMO results (directly derived from the DFT calculations) correspond to isolated oligomers in the gas phase and are therefore naturally referenced to the vacuum level. As a consequence, the absolute values differ approximately by the work function of Au(111), which is about 5.2–5.3 eV, in addition to any interface-induced renormalisation effects.

As an illustrative example, the AO monomer (A) exhibits STS resonances at –1.91 V (HOMO) and +1.11 V (LUMO) relative to the Fermi level, while the corresponding gas-phase HSE06 values are –6.04 eV and –3.78 eV relative to vacuum. After accounting for the Au(111) work function, the experimental levels correspond to approximately –7.2 eV and –4.2 eV on the vacuum scale, indicating stabilisation of about 1.2 eV and 0.4 eV, respectively, compared to the gas-phase HSE06 energies. Comparable shifts are obtained for the other oligomers listed in Table S2. This

degree of stabilisation is entirely consistent with expectations for molecules adsorbed on a polarisable metal substrate. Image-charge screening, interface dipoles and possible charge transfer are well known to lower the energies of molecular cationic and anionic states and to reduce frontier energy gaps by approximately 1–2 eV for conjugated molecules on metal or graphite surfaces.”

In principle, DFT calculations including an explicit Au(111) slab could be performed for a short nanoribbon. However, such calculations would require very large supercells to accommodate both the nanoribbon and the Au(111) herringbone reconstruction, and quantitative level alignment would in any case generally require many-body GW corrections beyond standard DFT. Moreover, the precise adsorption geometry, coverage and local environment are not uniquely determined experimentally, meaning that any single model would represent only one of many possible alignments. In view of these considerations, and given the good agreement already obtained for the gaps and their trends, we consider such calculations to be beyond the scope of the present work.

Finally, we fully agree with the Reviewer that a reliable identification of the measured resonances should rely on simulated STM/STS images.

12. *Which β values are chosen for equations 1 to 4? β_A and β_D are not defined further and cannot be calculated in the same way as the mixed β values as the Δ would be zero in this case, which would lead to no change in the energy levels of the homo dimers compared to the monomers.*

We thank the Reviewer for pointing out that we had not explicitly stated the values of the resonance integrals in the special case of the homodimers, and we apologise for this oversight. As correctly noted by the Reviewer, their definition cannot follow that used for the other oligomers, since the reference energy levels are degenerate in this case. In fact, the expressions for the variational corrections are also different here, as shown in Equations (1)–(4) of the SI. In line with our general aim of keeping the model as simple as possible, we have assumed a single constant value for β , taken to be 0.2 eV, consistent with the value used for the other oligomers. This has now been stated explicitly in the revised version of the SI.

13. *In Figure S21c one can recognize that the HOMO of DA lies higher in energy than the HOMO of D while the LUMO of DA lies below the LUMO of D. When combining DA and D to form DAD by applying Equations 5 and 6, which one is treated as the donor and which one as the acceptor? It appears that this choice should make a difference as Equations 5 and 6 are not symmetric with respect to swapping D and A. The same question arises for the combination of A and DA to form ADA in Figure S21b.*

In a general LCMO framework, the interaction between two molecular orbitals of compatible symmetry gives rise to two new hybridised orbitals: a lower-energy bonding combination and a higher-energy antibonding combination. When this concept is applied to any of the nanoribbon fragments, the two original LUMOs combine to form two new orbitals, with the bonding combination corresponding to the LUMO of the resulting oligomer. Similarly, the two original HOMOs combine to form two new orbitals, with the antibonding combination corresponding to the HOMO of the oligomer. Equations (5) and (6) describe this process for the DA heterodimer, but the same principle applies to all other fragment combinations, including DAD (formed from DA and D) and ADA (formed from A and DA), as schematically shown in Fig. S21.

14. *At the bottom of page 21 and the top of page 22 of the SI, the authors discuss that the results of the LCMO model do not depend on the order of monomers within a trimer but might depend on the choice of monomer and dimer a specific trimer is constructed from. To clarify this, the authors might elaborate on their choice of the specific orders and combinations presented in Table S2.*

Table S2 lists all combinations of monomers and dimers that yield distinct HOMO and LUMO energy levels within the extremely simplified LCMO framework presented here. The composition of each trimer is indicated by the parentheses defining the fragments used in its construction and the values reported in Fig. S23c correspond to the explicitly indicated combinations. The two trimers in Fig. S23c could, in principle, be represented through alternative fragment combinations: AAD could result from either the chosen (A)(AD) or the alternative (AA)(D) combination, and DDA from either the chosen (D)(DA) or the alternative (DD)(A) combination.

As discussed in the SI, the energy values obtained from these alternative constructions are not identical but differ only marginally (by a few percent). The specific combinations represented in Fig. S23c are thus effectively equivalent to the possible alternatives and not significant for our purposes, since—consistent with the aim of this simple model—we are primarily interested in the overall qualitative trends, which remain entirely unaffected by such minor differences.

Prompted by this helpful comment, we have revised the corresponding part of the SI to make this point more explicit. It now reads:

“So, for example, a DDA trimer can be obtained, as we have seen above, by combining a D monomer with a DA dimer as (D)(DA) (Fig. S21(c)). However, the same trimer could also result from joining a DD dimer with an A monomer as (DD)(A) (Fig. S22(b)). As is shown in Table S2, these two different paths result in slightly different values for the HOMO and LUMO energies. However, this is effectively irrelevant when looking only for qualitative trends, since the energetic differences are extremely small (within a few percent).”

15. *I would like to suggest to include some additional information to support the reader in following the derivation of the LCMO model with this and the following points. For Ref. 16 in the SI, a more precise location in the book, at least the chapter, could be mentioned.*

We thank the Reviewer for this suggestion and, in order to better guide the reader through the derivation of the LCMO model, we have now added the specific chapter of the book cited as Ref. 16 in the revised version of the SI.

16. *Under Equation 6 the nomenclature A' appears without a definition and can be found multiple times further down in the text. What is the significance of the prime?*

We thank the Reviewer for pointing out this potential source of confusion. The symbol the Reviewer refers to was not intended to denote a prime but rather a comma separating the different definitions of the terms used in Equations (5) and (6). To avoid any possible misunderstanding, we have now replaced these commas with semicolons in the revised version of the SI.

17. *Why, under which circumstances, is it justified to neglect the overlap integrals?*

In our elementary LCMO model (Section 21 of the SI), the donor and acceptor frontier orbitals are used as a minimal basis to construct the HOMO and LUMO of dimers and trimers. In a fully general LCMO treatment, one would indeed solve a generalised eigenvalue problem involving

both the Hamiltonian and overlap matrices. In the SI, however, we follow standard practice in qualitative MO and Hückel-type treatments and adopt the zero-overlap approximation, i.e. the off-diagonal overlap integrals are set to zero.

Formally, starting from a non-orthogonal set of monomer frontier orbitals, one may construct a Löwdin-orthogonalised basis set [Fischer-Hjalmar, I. Deduction of the Zero Differential Overlap Approximation from an Orthogonal Atomic Orbital Basis. *J. Chem. Phys.* 42, 1962–1972 (1965)]. In this orthogonal basis, the overlap matrix becomes the identity, while the Hamiltonian acquires renormalised diagonal (on-site) and off-diagonal (coupling) matrix elements. Importantly, the relative on-site energies are affected only at second order in the overlap, whereas the dominant effect of the orthogonalisation is a renormalisation of the effective couplings [Fischer-Hjalmar, I. Zero Differential Overlap in ϕ -Electron Theories. in *Advances in Quantum Chemistry* vol. 2 25–46 (Academic Press, 1966)]. In our LCMO model, we therefore work directly within an orthogonalised frontier-orbital picture, using phenomenological β parameters to represent these effective couplings and retaining the monomer HSE06 frontier energies as on-site terms.

We emphasise that this simplified LCMO treatment is intended solely to rationalise qualitative trends in frontier orbital energies as a function of composition and sequence, rather than to provide quantitatively accurate level positions. This is evident in Fig. S23, where the LCMO results capture the overall ordering and evolution of the HOMO, LUMO and gap values observed in both DFT and STS, while deviations remain for more subtle cases—such as the additional gap narrowing in block-type trimers—where a simple zero-overlap model is expected to break down.

Identification of products with bond-resolved STM or AFM

18. Whereas the different appearance of the O-C-O and C=O groups is quite clear for the pure ribbons in Figure 2c,g, this is harder to recognize for the mixed ribbons in Figure 4. For example, the bottom AO unit in Figure 4d only has a clear C=O group on the right side but not on the left side. Additionally, the two sides in the bottom unit in Figure 4f and the top two units in Figure 4h appear different from each other and are not clearly distinguishable from the O-C-O in these images, at least not in the chosen contrast. The identification should be supported further. Can the authors exclude that the left and right side of a unit might be different, e.g. that a keto group has been lost during or after the synthesis?

We agree with the Reviewer that the two sides of a nanoribbon unit can sometimes exhibit a slightly different appearance, even when they terminate with the same functional group. However, we maintain that the two functional groups considered here still retain distinct and reproducible characteristics. In both nc-AFM and BR-STM images, the C=O group consistently appears with a sharp, well-defined protrusion, whereas the O–C–O edge does not and instead exhibits a darker, more diffuse ring-like contrast.

We believe that most differences in bond-resolving images usually relate to the presence of other locally adsorbed nanoribbons or Br atoms. In particular, a darker attractive region is measured in the nc-AFM images in the regions between ribbons, resulting in one side of a nanoribbon with a slightly different appearance to the other when adsorbed at the edge of a molecular island. Asymmetry in the tip can also have this effect.

On this basis, we find no evidence that the left and right sides of a given unit correspond to chemically different terminations, nor that a keto group has been lost during or after synthesis.

19. *In line with the previous point, I wonder whether a direct comparison of the same BR imaging method (STM or AFM) of both precursors and the corresponding final ribbons might be useful to further support the identification of the O-C-O and C=O groups. In the image of Br2PXX presented in S25 the appearance of the oxygen does not seem unambiguous.*

We thank the Reviewer for this suggestion. In response to this comment, and to the preceding one, we have added a new figure to the SI (Fig. S27). This figure shows an AO molecule adsorbed alongside a PXX molecule, with the simultaneously recorded tunnelling current and frequency-shift signals, as well as the Laplace-filtered frequency-shift image, presented in separate panels.

We believe that this direct, side-by-side comparison will be helpful to clearly demonstrate that the two units can be reliably distinguished when considering the appearance of the functional groups across the different signal channels. The figure also illustrates how the contrast can vary when only one side of a molecule is adsorbed adjacent to another molecule or nanoribbon, further rationalising the asymmetries discussed above.

20. *In Figure 2c and Figure 4f,h the benzene rings next to those rings containing the C-OC group appear quite distorted. In comparison those rings next to rings with the C=O groups appear less distorted, e.g. in Figure 2g and Figure 4h (except in Figure 4f). What could be the reason for this?*

We agree with the Reviewer that differences in the apparent distortion of the benzene rings can be observed when comparing images of PXX and AO nanoribbons (Fig. 2). We attribute these distortions primarily to differences in the relaxation and lateral motion of the CO-functionalised tip during imaging, which is well known to influence the resulting contrast in bond-resolved AFM and STM images. In particular, the CO tip is likely to experience different force gradients when passing over or near rings containing the oxygen atoms of the PXX units, leading to enhanced apparent distortions in those cases.

For the mixed nanoribbons shown in Fig. 4, some of the observed distortions are more likely related to the specific tip structure. As discussed in the response to comment 1 and in Section 24 of the SI, tip 1 in particular produces more distorted images overall, which also explains why the contrast in Fig. 4(f) differs from that in Fig. 4(h).

21. *In Figure S7b, a specific brightness in the bond-resolved STM is attributed to the shift of the HOMO to higher energies. However, it appears that the brightest area has a well-defined hexagonal shape with sharp corners usually indicate of geometric-type contrast. (On the other hand, a brightness variation in the longer ribbon in Figure S17 is interpreted as due to the topography of the herringbone of Au(111), not due to electronic structure.) These brightness variations might be relevant for the discussions raised in points 6. and 7 above. Experimentally, one could confirm whether the brightness is due to electronic structure or geometric structure (e.g. adsorption geometry) by using even smaller bias voltages or larger tip-sample distances.*

The Reviewer is correct that the contrast observed when imaging longer PXX nanoribbons reflects a combination of geometric and electronic effects. In particular, the hexagonal features with sharp corners arise from the CO-functionalised tip entering the repulsive regime, while superimposed variations in the tunnelling current reflect changes in the local density of states of the molecule. This mixed contrast is observed irrespective of whether the ribbons are imaged at +40 mV or +5 mV and is already noted in the caption of Fig. S7.

On the other hand, the brightness variations observed for the long nanoribbon shown in Fig. S17 originate solely from electronic effects associated with the nanoribbon and the underlying Au(111) herringbone reconstruction. In this case, the tip-sample distance was significantly larger than in the bond-resolved STM images, such that repulsive interactions between the CO tip and the nanoribbon did not contribute to the contrast. These images therefore serve as a useful comparison with the BR-STM images in Figs. S7(b) and S7(c), which were recorded with the CO tip positioned much closer to the molecule.

Variations in contrast arising from the underlying herringbone reconstruction can be neglected for the short and mixed nanoribbons. The short ribbons preferentially adsorb in between herringbone lines, while the mixed nanoribbons are located within or adjacent to islands of nanoribbons and Br atoms, where the Au(111) herringbone reconstruction is lifted due to Br chemisorption.

Nevertheless, subtle effects of the underlying herringbone reconstruction can be observed in the nc-AFM image of the PXX nanoribbon shown in Fig. 2(c). Motivated by the Reviewer's observation, we have added the following sentence to the caption of Fig. 2 to make this explicit:

"The brighter upper and lower sections of the nanoribbon are related to the position of the ribbon relative to the underlying Au(111) herringbone reconstruction."

22. *Is the structure overlaid in Figure S10d a DFT result or are the molecules manually placed on top of the image? In the latter case, which modelling do the authors refer to in line 180 of the main text ("supported by modelling")?*

We thank the Reviewer for raising this point, which has given us the opportunity to clarify the text in the revised version of the paper. While the structures of the individual VO₃ molecules in Figure S10d were geometry-optimised by DFT, the overlay onto the BRSTM images was performed manually to achieve the best agreement with the positions and orientations of the bonds resolved in the measurements. However, we agree that the term "modelling" may inadvertently suggest that the overlaid structures were obtained through computational simulations. To avoid any potential confusion, we have replaced "modelling" with "structural assignment" in the revised manuscript.

Synthesis as observed with STM

23. *In Figure 2e it is hard to recognize the molecules because the image contrast is divided between at least three terraces. Is it a full monolayer coverage or is the orange area in the center a molecular island and the red area next to it is the bare surface? The image in Figure S11d might provide more useful information in this place.*

We thank the Reviewer for pointing out that there could be some unclarity in the interpretation of Fig. 2(e). The coverage of the VO₃ molecules in this image is close to a full monolayer. The brighter region in the centre corresponds to a molecular island, while the adjacent, slightly dimmer region represents the remaining bare portion of the same terrace extending towards the step edge. A bare, herringbone-reconstructed region of the Au(111) surface is visible in the upper part of the image.

To clarify this, we have added a reference to Fig. S10 in the revised manuscript and included the following sentence in its caption:

“The molecular coverage on the large terrace at the centre of the image is close to a full monolayer, with a region of bare Au(111) visible in the upper part of the image.”

24. In lines 181-182 of the main text the authors claim that annealing above 473 K already results in NRs with average length. However, in Figure S11b,e mainly 3-fold symmetric structures contained in islands can be recognized, whereas S11f (after subsequent higher annealing) then shows longer NR features. The change from S11e to S11f cannot be explained by the mere self-assembly enabled by Br desorption as claimed by the authors, but rather further reaction(s) appear to proceed here. This could be discussed in line with literature finding the reaction to fully proceed only after Br desorption (see my comment 29 below and Refs. 4,5 of the SI).

We thank the Reviewer for pointing this out and agree that the description in the original version was not sufficiently clear. We have therefore revised the manuscript accordingly. The text now reads:

“The polymerisation behaviour of VO₃ closely resembles that of Br₂PXX. Annealing to 473 K results in the formation of short nanoribbons that are co-adsorbed alongside bromine atoms. Further annealing to 573 K leads to bromine desorption (Fig. S11 and Fig. 2(f)) and the formation of long nanoribbons. Unlike PXX nanoribbons, the AO nanoribbons tend to assemble into compact domains following bromine desorption. A tentative molecular arrangement for this assembly, based on weak hydrogen bonding between ketone groups and adjacent hydrogens, is presented in Fig. S12.”

25. The authors state that Br atoms are present on the surface in Figure S3b (line 122) (and in Figure S11b,e). How and where can the reader recognize the Br atoms in the STM images, specifically in comparison to Figure S3c where they are claimed to be desorbed? This could be described in words or marked in the image and the image quality in the pdf of the SI might be improved to aid the reader in recognizing the Br atoms.

We thank the Reviewer for raising this point. The presence of Br atoms in Fig. S3(b) is inferred indirectly from the self-assembly behaviour of the nanoribbons, which form mixed nanoribbon–bromine islands when Br is present on the surface. As discussed in Section 3 of the SI, the sample shown in Fig. S3(c) was annealed to 523 K, at which point some (but not all) of the bromine has desorbed. This is evidenced by the more dispersed distribution of the nanoribbons compared to Fig. S3(b). In the text, this behaviour is further contrasted with the lower-coverage sample shown in Fig. 2(b) of the main text, where almost all bromine has desorbed at the same temperature. In that case, the PXX nanoribbons are generally spread across the surface, as they do not self-assemble in the absence of co-adsorbed bromine. This behaviour is distinctly different from that of the AO nanoribbons, which are observed to self-assemble even after bromine desorption.

To clarify this point further, we have added the following sentence to the main text:

“This also results in the dispersion of the PXX nanoribbons across the surface, as they do not self-assemble without co-adsorbed bromine atoms.”

26. In Line 137 the authors state “STM images reveal that many nanoribbons exhibit structural defects”. However, Figure S5 referred to in the same paragraph does not contain STM images and the images in Figure S6 are not completely convincing in comparison to the given structures. The presence of the 5-membered rings might be confirmed with bond-resolved STM or AFM.

We thank the Reviewer for this comment and for the opportunity to clarify this point. Fig. S5 is referenced in the subsequent sentence of the same paragraph and is included to discuss the structure of possible contaminant species, rather than to provide STM evidence of defects in the nanoribbons. To make this clearer, we have revised the text as follows:

“For instance, molecules brominated at the 1-position instead of the 3-position (1 in Fig. S5) tend to form ‘straight’ rather than the regular ‘alternating’ junctions, while tribrominated species (2 in Fig. S5) may result in branched structures, similar to those observed at the centre of Fig. S6b, highlighted by the red circle.”

With regard to Fig. S6, we consider the STM images shown there to provide reasonable evidence for the presence of five-membered rings. In particular, the characteristic deviations in angle, shape and registry clearly visible in these images cannot be readily explained by simple conformational variations of the regular nanoribbon backbone and are instead only consistent with the incorporation of five-membered rings.

27. Regarding the bottom part of Table S1: What are the “monomer units”, precursor or monomer products or both? Why is the product PXX (25 units) compared to the precursor VO3 (166 units)? How do these numbers lead to the percentages of 43 % and 57 % between PXX and AO?

We thank the Reviewer for pointing out that the interpretation of Table S1 required clarification, which has given us the opportunity to improve it. First, the label “VO3” in the table was incorrect and has now been replaced by the correct label, “AO”, for which we apologise. We also agree that the term “alone” was insufficiently clear. In our original intent, this term referred to individual PXX or AO units observed on the surface that are not incorporated into nanoribbons, irrespective of whether bromine atoms remain attached. To make this explicit, we have replaced “alone” with “isolated units (not in nanoribbons)” and have clearly labelled the bottom section of Table S1 as “Total constituent units observed (ribbons + isolated)”.

The percentages reported at the bottom of the table correspond to the total number of each type of constituent unit (PXX or AO) observed in a given experiment, including both units incorporated into nanoribbons and isolated units. These totals are then used to estimate the relative abundance of PXX and AO units after annealing and to compare this with the observed junction statistics.

We have revised the text in Section 18 of the SI to make this distinction explicit. The revised text now reads:

“A number of junctions were examined via BR-STM imaging for the mixed Br₂PXX and VO3 deposition and annealing experiments. The results of the statistical analysis are reported in Table S1. For each sample, the table lists the percentage of each type of junction/connection observed. The number of individual units remaining on the surface but not incorporated into nanoribbons is also reported, followed by the total percentages of each type of constituent unit (PXX or AO) observed for that sample, including both ribbon-incorporated and isolated units.

This analysis reveals that the monomers show a greater tendency to form D–A connections rather than D–D or A–A connections. This effect is particularly pronounced for AO, which is present in excess in both the low- and high-coverage experiments.”

XPS analysis

28. On page 5 of the SI the authors write “A gradual transition to lower binding energies is observed between 413 K and 453 K and is interpreted as the result of the transition from precursor to nanoribbon in a similar manner to that seen in other studies.^{4,5}” In both references 4 and 5 mentioned here, the main C 1s peak shifts first to lower and then again to higher binding energies in the course of the reaction. The authors of these articles distinguish between different reaction steps and a C-Au bond is included in the reaction paths in both. Which step is explained here by the transition to lower binding energies and how can an intermediate containing a C-Au bond be excluded based on the XPS results?

Please see answer to comment 29.

29. In particular, Ref. 4 in the SI (Batra et al., Chem. Sci., 2014, 5, 4419) argues that the reaction to the GNR only occurs after full removal of the Br from the surface and interprets the initial shift in the C 1s binding energy as the formation of an intermediate containing C-Au. The Br 3d binding energies as well as the desorption temperature of Br from Au(111) observed here appear to agree with this literature, however, the C 1s shifts do not (see previous comment). How can the comparison between these studies lead to the conclusion that the reaction has fully happened? How can the authors also exclude other possible reasons for shifts in the C 1s peaks under annealing like a change in the work function (see page 5 of the SI). Another signal like the Au 4f might be checked for comparison.

We thank the Reviewer for these detailed and thoughtful comments on the XPS results presented in the SI. We first emphasise that the XPS data are intended as supporting evidence for the STM-based observations, rather than as the primary means of tracking reaction progress.

Across all nanoribbon syntheses discussed in this work, we clearly observe the coexistence of nanoribbons and bromine atoms on the surface, which indicates that complete bromine desorption is not a prerequisite for nanoribbon formation. In particular, the mixed sample shown in Fig. 4 of the main text retains a substantial amount of bromine, while the SI figures for the growth of the pure nanoribbons also show nanoribbons coexisting with bromine at lower annealing temperatures. Although bromine desorption coincides with the formation of longer AO nanoribbons at higher temperatures, it is not clear whether this is causally linked to the growth process or simply reflects the higher temperatures required for their formation.

The Reviewer is correct in noting the ambiguity in assigning the origin of the C 1s binding-energy shift observed between approximately 413 K and 473 K. As discussed in the cited literature, this temperature range likely encompasses multiple processes occurring concurrently. While the formation of C–Au–bonded intermediates has been proposed in previous studies, we did not observe any clear organometallic intermediate in our STM experiments, and therefore we cannot assign the C 1s shift unambiguously to such a step. At the same time, we cannot exclude that transient intermediates of this type occur during annealing but escape direct experimental observation.

With respect to bromine desorption and its influence on the C 1s binding energies via work-function changes, we do not observe a pronounced shift in the C 1s peak that could be directly attributed to Br desorption. Prompted by the Reviewer’s comments, a closer inspection suggests that a very small upshift at temperatures above 473 K may indeed be present in the C 1s spectra shown in Fig. S3d. However, given the limited magnitude of this effect, and the absence of a clear step-like behaviour, we refrain from assigning it a specific microscopic origin.

To reflect these ambiguities more accurately, we have revised the text in the SI as follows:

“A gradual transition to lower binding energies is observed between 413 K and 453 K and is interpreted as the result of the transition from precursor to nanoribbon in a similar manner to that seen in other studies.^{4,5} We cannot rule out that part of this transition is related to the formation of intermediate species, such as organometallic C–Au–bonded structures. However, if such intermediates occur, they do not appear as a distinct reaction step that can be clearly identified in corresponding STM experiments. We also do not observe a pronounced shift in the C 1s binding energy upon bromine desorption that would be expected from a significant work-function change, although a subtle upshift may be present at temperatures above 473 K in Fig. S3d.”

30. In lines 120-122 the authors specifically claim that the XPS results indicate the formation of the desired ultra-narrow nanoribbons. Whereas the XPS does indicate a reaction, specifically the removal of Br and some change(s) in C bonds, this is not sufficient to prove that exactly this product has formed. This claim can far more easily be made with the bond-resolved SPM.

We thank the Reviewer for this comment and agree with their assessment. While the XPS data clearly indicate that a reaction takes place—most notably debromination and changes in the carbon bonding environment—they do not, on their own, uniquely identify the formation of the ultra-narrow nanoribbons. As the Reviewer correctly points out, this structural identification is most convincingly established by the bond-resolved SPM measurements.

In light of this comment, and in line with our responses to the previous comments regarding the interpretation of the XPS data, we have toned down the corresponding statement in the main text to emphasise the supportive role of the XPS measurements rather than treating them as definitive proof of nanoribbon formation. The formation and atomic structure of the ultra-narrow nanoribbons are therefore attributed primarily to the STM and bond-resolved SPM results, with XPS providing complementary information on the thermal evolution of the system.

The revised version of this sentence now reads:

“X-ray photoelectron spectroscopy (XPS) measurements performed during the nanoribbon growth process (Fig. S3(d)) indicate the onset of debromination and changes in the carbon bonding environment upon annealing above approximately 410 K, providing supporting evidence for the nanoribbon formation observed by STM.”

31. In line with the previous points, the reaction steps, Ullmann coupling and subsequent dehydrogenation, could be made clearer to the more general reader e.g. in Figure 1 of the main text.

We thank the Reviewer for this suggestion, but respectfully do not think that this is necessary, as we believe that Fig. 2 of the main text sufficiently conveys the nature of the final products to a general reader. Further details of the reaction are intentionally relegated to the SI, together with discussion of the relevant literature, since this is not the main focus of the paper.

32. The caption to Figure S3 states “These spectra demonstrate the debromination of Br₂PXX between 453 K and 493 K.” The debromination of Br₂PXX seems to be halfway processed at 453 K already. Did the authors test lower annealing temperatures as well?

We thank the Reviewer for pointing this out and apologise for the imprecise wording in the original caption. The Reviewer is correct that the debromination of Br₂PXX has already begun by 453 K and therefore most likely starts between 413 K and 453 K. We have corrected the caption to Fig. S3 accordingly. The revised caption now reads:

“These spectra demonstrate that the debromination of Br₂PXX has started between 413 K and 453 K (forming a chemisorbed Au-Br species with a different binding energy) and that the bromine atoms have fully desorbed after annealing to 553 K.”

Other comments

33. Lines 161-175 of the manuscript contain the same text as lines 137-151.

We are very grateful to the Reviewer for pointing out this error and apologise for the oversight, which has now been corrected.

34. The resolution of the figures in the pdf should be improved to allow the reader to properly recognize e.g. the relevant features of the SPM images.

We have made efforts to improve the resolution of the most critical figures in the revised manuscript. However, we believe that this issue may partly arise from the file conversion used during the editorial review process. We expect that the image quality will be further improved in the final published version.

Reviewer report to manuscript “Ultra-narrow donor-acceptor nanoribbons”

J. Lawrence *et al.* employ monomer precursors of donor and acceptor type to synthesize nanoribbons of pure as well as mixed character on a gold surface. The authors follow the synthesis with scanning tunneling microscopy and X-ray photoelectron spectroscopy and claim the identity of the products based on bond-resolved STM or AFM. They further discuss the influence of length and composition on the band gaps of these ribbons as measured by scanning tunneling spectroscopy in comparison to density functional theory and aim to develop a model based on the linear combination of orbitals to comprehend and predict these trends.

The goal of the manuscript, translating the combination of donor and acceptor monomers from solution to on-surface synthesis and thereby improving the atomic-scale precision of tuning band gaps, pertains to a relevant topic and appears to be a promising approach, which is clearly described in the introduction. However, while I appreciate the high-resolution STS, bond-resolved imaging and the amount of measurements that were performed, significant improvements in the analysis, interpretation, and presentation of the data are needed to support the claims and to warrant publication. My major concerns are that a reliable experimental identification of HOMO and LUMO of all investigated species, which is crucial for all discussion of the measured gaps, is currently lacking and that a significant mathematical issue is present in the simplified LCMO model and the conclusions drawn from it, as well as that specific details of the synthesis as investigated with STM and XPS remain unclear.

Generally, the current manuscript is challenging to read as many pieces of information relevant for following a small part of the main text are distributed throughout the extensive SI without a clear order to help the reader. This opens up questions about the focus of the paper. Is it the detailed understanding of the synthesis including a discussion of the possible (by-)products and coupling mechanisms or the detailed analysis and theoretical understanding of the electronic structure as suggested by the nice introduction? Whereas I find the synthesis in itself highly interesting as well and agree that it is a necessary prerequisite for all further studies of the products, the extensive investigation the authors present regarding both topics does not appear to fit in a concise way into this one communication.

I hope my detailed questions and comments listed below, sorted by topics, may support the authors in improving their work. The line numbers mentioned here refer to the pdf document of the main text, whereas the positions in the SI are referred to by specific page numbers or figures.

Identification of orbitals in STS and dI/dV images

1. In Figure 3, Figure 4, Figure S15, and Figure S26 the authors compare dI/dV images taken at the specific voltages where orbitals, in most cases HOMO and LUMO, are expected from the STS and compare them to visualizations of DFT orbitals in order to

confirm the energetic position of said orbitals (e.g. lines 209-212). Generally, this comparison is challenging to do directly by eye, especially due to the p-wave contribution which the authors mention and the fact that the CO-tip can lead to even more complex contrasts due to an overlap of both p and s contributions. Which one dominates depends for example on the tip-sample distance. A significantly more convincing agreement could be achieved easily by showing a direct comparison between the measured dI/dV images and simulated STM images, which is also the common approach in the literature. As the authors have already calculated the DFT orbitals, such simulations can be realized with little computational effort. A further advantage is that such simulations can be done assuming both an s- or p-wave tip and each measured image might be reproduced well by combining a specific ratio of s and p contributions in the simulation.

2. In line with the previous point, I have to disagree in particular with the statement: “shows excellent agreement in both shape and spatial distribution, considering the p-wave character of the CO tip” in line 311 of the main text. Looking e.g. at the dI/dV images of the HOMO in Figure 4g and the LUMO in Figure 4h, I cannot recognize any p-wave contribution. The p-wave character should lead to a different number and different positions of nodes in the STM contrast compared to the appearance of an s-wave image and thus the appearance of the DFT orbital. This can be understood as p-wave images can be modeled employing the lateral derivative of the wave function (Gross *et al.*, PRL 107, 086101 (2011), doi.org/10.1103/PhysRevLett.107.086101) as opposed to the wave function itself as is done for s-wave simulations. Additionally, the comparison between a metallic and a CO tip provided in Figure S15 b along with the description “generally similar for both tips, with finer detail resolved by the CO tip.” might rather suggest that the CO tip did not lead to exclusively p-wave tunneling here, but this can only be judged clearly by showing STM simulations.
3. In the STS presented in Figure 3a, how can one recognize the marked position of the LUMO for the PXX dimer? There appears to not be an obvious peak there.
4. The LUMO for the PXX monomer appears to be outside the measured bias voltage range in Figure 3a and no value is given in Table S2. As changes of the HOMO and LUMO positions with length and with composition of the ribbons are the main topic of the manuscript, this would be a crucial point of comparison and should be measured and discussed in comparison to the theory.
5. In Figure 4c, the spectrum for PXX-AO exhibits some smaller peaks around -0.1 eV and +0.1 eV. What is their origin? Similarly, the orange curve in Figure S25 shows an additional peak at -250 mV. Generally, for all spectra given in the main text (Figure 3a,f, and Figure 4c), a spectrum of the Au(111) surface for comparison might be useful.
6. In Figure S16 the broad HOMO peak appears clearly visible for the 2-mer and 3-mer. However, from the 4-mer on a significantly sharper peak appears (even sharper in b

due to the chosen oscillation). (This is surprising in comparison to the statement “the unoccupied resonances of PXX NRs are significantly broader than the occupied states” in lines 229-230.) The sharp peak then shifts further to higher energies with NR length but significantly more slowly than the initial shift of the HOMO peak from the 2-mer to the 3-mer. As most clearly visible for the 4-mer for both tips, the broader HOMO peak might still be present underneath the sharper one, see around 0.06 V in Figure a and around -0.05 V in Figure b. Therefore, the authors should discuss other possible origins for the sharp peak close to the Fermi energy and its shape. For example, acenes (which are basically ultrathin zigzag edge NRs) are predicted to have open shell character increasing with their length, which leads to Kondo features close to the Fermi energy that have been measured with STS (Ruan *et al.*, *J. Am. Chem. Soc.* 2025, 147, 4862–4870, doi.org/10.1021/jacs.4c13296) and can be fitted by appropriate functions (Fano or Frota line shapes).

7. From Figures S14 and S20 (showing in the standard STM contrast, which easily appears wider than the geometry of the ribbon itself) and Figure S24 where the points are drawn next to the resolved bonds, it appears that all STS spectra presented are measured at the edges of the ribbons. I appreciate that the filled and empty states might be measured at separate positions depending on the spatial distributions of HOMO and LUMO, but the positions all appear deliberately quite far on the outside of the ribbons, which should be explained. Generally, comparing STS at different positions in one and the same ribbon might be useful for a more complete picture, especially in the center compared to the edges and at the different sections of the mixed ribbons. Specifically, at the edges complex and interesting electronic features like edge states might be present (Wang *et al.*, *Nat Commun* 7, 11507 (2016). doi.org/10.1038/ncomms11507).
8. In Figure S14, it could aid the reader if the orientation of each structure was chosen the same as in the STM image (e.g. the structures in the top row second from the left and on the right would need to be mirrored).
9. Line 274-276: HOMO and LUMO levels are described as “downshifted” and “upshifted”: With respect to the initial position of which ribbon/monomer is this shift?

LCMO model and comparison of theoretical to measured band gaps

10. In lines 280-281 of the main text the authors claim that the LCMO model contains “variational corrections that scale inversely with the energy spacing”. Similar and more detailed descriptions are found in the SI, e.g. “This is because of the large values of ΔE_{HOMO} and ΔE_{LUMO} in equations (5) and (6), resulting in rather small corrections to E_{HOMOD} and E_{LUMOA} , respectively.” and “because of the small value of ΔE_{HOMO} at the denominator in equation (5).” I need to disagree with these statements and the conclusions drawn from them in the main text and the SI (pages 20 to 22). Whereas in Equations 5 and 6 it does appear that the correction factor decreases with the Δ as

it appears in the denominator, the assumption that β is proportional to Δ leads to a Δ^2 in the numerator, thus leaving a Δ in the numerator (instead of the denominator) after reducing the fraction. Thus, the correction factor mathematically speaking actually increases with Δ . This appears to be compensated by choosing a smaller proportionality factor (the 2, 0.5, or 0.2 at the end of page 19 of the SI) within the β when the Δ is larger, such that in total a smaller correction factor still results for larger Δ . However, the origin of the proportionality factor and why it should be smaller for larger Δ remains unclear. The authors give the information that these factors appear appropriate “a posteriori” because the results agree with those from DFT. Overall, the model thus appears more like a fitting procedure with the proportionality factors being the resulting fit parameters that describe the measured trends rather than being a theoretical model from which conclusions could be drawn in comparison to DFT results as is currently claimed.

11. In Table S2, the authors report absolute values for HOMO and LUMO energies as well as resulting gaps comparing the measurement and the DFT as well as the LCMO model. The absolute energetic positions of the measured HOMO and LUMO largely deviate from the results of DFT (and the LCMO model), which appears to not be clearly mentioned or discussed anywhere in the text. Whereas it is well-known that the absolute energies from DFT often vary from experimental ones, deviations between 4 and 5 eV seem rather significant. For example, the role of the surface (which is mentioned by the authors in line 294 in the context of the gaps) should be discussed in the context of the absolute energy levels here as well. Literature examples of such large energy level changes due to adsorption on a surface might support the idea or the authors could perform a DFT calculation including the surface on just one short example ribbon to test this. Due to these large deviations, a reliable identification of the specific measured orbitals by means of simulated STM images (as mentioned above) is even more crucial.
12. Which β values are chosen for equations 1 to 4? β_A and β_D are not defined further and cannot be calculated in the same way as the mixed β values as the Δ would be zero in this case, which would lead to no change in the energy levels of the homo dimers compared to the monomers.
13. In Figure S21c one can recognize that the HOMO of DA lies higher in energy than the HOMO of D while the LUMO of DA lies below the LUMO of D. When combining DA and D to form DAD by applying Equations 5 and 6, which one is treated as the donor and which one as the acceptor? It appears that this choice should make a difference as Equations 5 and 6 are not symmetric with respect to swapping D and A. The same question arises for the combination of A and DA to form ADA in Figure S21b.
14. At the bottom of page 21 and the top of page 22 of the SI, the authors discuss that the results of the LCMO model do not depend on the order of monomers within a trimer but might depend on the choice of monomer and dimer a specific trimer is

constructed from. To clarify this, the authors might elaborate on their choice of the specific orders and combinations presented in Table S2.

15. I would like to suggest to include some additional information to support the reader in following the derivation of the LCMO model with this and the following points. For Ref. 16 in the SI, a more precise location in the book, at least the chapter, could be mentioned.
16. Under Equation 6 the nomenclature A' appears without a definition and can be found multiple times further down in the text. What is the significance of the prime?
17. Why, under which circumstances, is it justified to neglect the overlap integrals?

Identification of products with bond-resolved STM or AFM

18. Whereas the different appearance of the O-C-O and C=O groups is quite clear for the pure ribbons in Figure 2c,g, this is harder to recognize for the mixed ribbons in Figure 4. For example, the bottom AO unit in Figure 4d only has a clear C=O group on the right side but not on the left side. Additionally, the two sides in the bottom unit in Figure 4f and the top two units in Figure 4h appear different from each other and are not clearly distinguishable from the O-C-O in these images, at least not in the chosen contrast. The identification should be supported further. Can the authors exclude that the left and right side of a unit might be different, e.g. that a keto group has been lost during or after the synthesis?
19. In line with the previous point, I wonder whether a direct comparison of the same BR imaging method (STM or AFM) of both precursors and the corresponding final ribbons might be useful to further support the identification of the O-C-O and C=O groups. In the image of Br₂PXX presented in S25 the appearance of the oxygen does not seem unambiguous.
20. In Figure 2c and Figure 4f,h the benzene rings next to those rings containing the C-O-C group appear quite distorted. In comparison those rings next to rings with the C=O groups appear less distorted, e.g. in Figure 2g and Figure 4h (except in Figure 4f). What could be the reason for this?
21. In Figure S7b, a specific brightness in the bond-resolved STM is attributed to the shift of the HOMO to higher energies. However, it appears that the brightest area has a well-defined hexagonal shape with sharp corners usually indicate of geometric-type contrast. (On the other hand, a brightness variation in the longer ribbon in Figure S17 is interpreted as due to the topography of the herringbone of Au(111), not due to electronic structure.) These brightness variations might be relevant for the discussions raised in points 6. and 7 above. Experimentally, one could confirm whether the brightness is due to electronic structure or geometric structure (e.g. adsorption geometry) by using even smaller bias voltages or larger tip-sample distances.

22. Is the structure overlaid in Figure S10d a DFT result or are the molecules manually placed on top of the image? In the latter case, which modelling do the authors refer to in line 180 of the main text (“supported by modelling”)?

Synthesis as observed with STM

23. In Figure 2e it is hard to recognize the molecules because the image contrast is divided between at least three terraces. Is it a full monolayer coverage or is the orange area in the center a molecular island and the red area next to it is the bare surface? The image in Figure S11d might provide more useful information in this place.
24. In lines 181-182 of the main text the authors claim that annealing above 473 K already results in NRs with average length. However, in Figure S11b,e mainly 3-fold symmetric structures contained in islands can be recognized, whereas S11f (after subsequent higher annealing) then shows longer NR features. The change from S11e to S11f cannot be explained by the mere self-assembly enabled by Br desorption as claimed by the authors, but rather further reaction(s) appear to proceed here. This could be discussed in line with literature finding the reaction to fully proceed only after Br desorption (see my comment 29 below and Refs. 4,5 of the SI).
25. The authors state that Br atoms are present on the surface in Figure S3b (line 122) (and in Figure S11b,e). How and where can the reader recognize the Br atoms in the STM images, specifically in comparison to Figure S3c where they are claimed to be desorbed? This could be described in words or marked in the image and the image quality in the pdf of the SI might be improved to aid the reader in recognizing the Br atoms.
26. In Line 137 the authors state “STM images reveal that many nanoribbons exhibit structural defects”. However, Figure S5 referred to in the same paragraph does not contain STM images and the images in Figure S6 are not completely convincing in comparison to the given structures. The presence of the 5-membered rings might be confirmed with bond-resolved STM or AFM.
27. Regarding the bottom part of Table S1: What are the “monomer units”, precursor or monomer products or both? Why is the product PXX (25 units) compared to the precursor VO3 (166 units)? How do these numbers lead to the percentages of 43 % and 57 % between PXX and AO?

XPS analysis

28. On page 5 of the SI the authors write “A gradual transition to lower binding energies is observed between 413 K and 453 K and is interpreted as the result of the transition from precursor to nanoribbon in a similar manner to that seen in other studies.^{4,5}” In both references 4 and 5 mentioned here, the main C 1s peak shifts first to lower and then again to higher binding energies in the course of the reaction. The authors of these articles distinguish between different reaction steps and a C-Au bond is

included in the reaction paths in both. Which step is explained here by the transition to lower binding energies and how can an intermediate containing a C-Au bond be excluded based on the XPS results?

29. In particular, Ref. 4 in the SI (Batra et al., Chem. Sci., 2014, 5, 4419) argues that the reaction to the GNR only occurs after full removal of the Br from the surface and interprets the initial shift in the C 1s binding energy as the formation of an intermediate containing C-Au. The Br 3d binding energies as well as the desorption temperature of Br from Au(111) observed here appear to agree with this literature, however, the C 1s shifts do not (see previous comment). How can the comparison between these studies lead to the conclusion that the reaction has fully happened? How can the authors also exclude other possible reasons for shifts in the C 1s peaks under annealing like a change in the work function (see page 5 of the SI). Another signal like the Au 4f might be checked for comparison.
30. In lines 120-122 the authors specifically claim that the XPS results indicate the formation of the desired ultra-narrow nanoribbons. Whereas the XPS does indicate a reaction, specifically the removal of Br and some change(s) in C bonds, this is not sufficient to prove that exactly this product has formed. This claim can far more easily be made with the bond-resolved SPM.
31. In line with the previous points, the reaction steps, Ullmann coupling and subsequent dehydrogenation, could be made clearer to the more general reader e.g. in Figure 1 of the main text.
32. The caption to Figure S3 states “These spectra demonstrate the debromination of Br₂PXX between 453 K and 493 K.” The debromination of Br₂PXX seems to be halfway processed at 453 K already. Did the authors test lower annealing temperatures as well?

Other comments

33. Lines 161-175 of the manuscript contain the same text as lines 137-151.
34. The resolution of the figures in the pdf should be improved to allow the reader to properly recognize e.g. the relevant features of the SPM images.